# WAVELET PREDICTIVE REPRESENTATIONS FOR NON-STATIONARY REINFORCEMENT LEARNING

**Min Wang**[1], **Xin Li**[1]*, **Ye He**[1], **Yao-Hui Li**[1], **Hasnaa Bennis**[1], **Riashat Islam**[2],
**Mingzhong Wang**[3]
[1] Beijing Institute of Technology, China     [2] Microsoft Research, AI Frontiers, New York City
[3] University of the Sunshine Coast, Australia
{minwangcs,xinli,heye,liyaohui,3820252027}@bit.edu.cn
riashatislam@microsoft.com     mwang@usc.edu.au

## ABSTRACT

The real world is inherently non-stationary, with ever-changing factors, such as weather conditions and traffic flows, making it challenging for agents to adapt to varying environmental dynamics. Non-Stationary Reinforcement Learning (NSRL) addresses this challenge by training agents to adapt rapidly to sequences of distinct Markov Decision Processes (MDPs). However, existing NSRL approaches often focus on tasks with regularly evolving patterns, leading to limited adaptability in highly dynamic settings. Inspired by the success of Wavelet analysis in time series modeling, specifically its ability to capture signal trends at multiple scales, we propose WISDOM to leverage wavelet-domain predictive task representations to enhance NSRL. WISDOM captures these multi-scale features in evolving MDP sequences by transforming task representation sequences into the wavelet domain, where wavelet coefficients represent both global trends and fine-grained variations of non-stationary changes. In addition to the auto-regressive modeling commonly employed in time series forecasting, we devise a wavelet temporal difference (TD) update operator to enhance tracking and prediction of MDP evolution. We theoretically prove the convergence of this operator and demonstrate policy improvement with wavelet task representations. Experiments on diverse benchmarks show that WISDOM significantly outperforms existing baselines in both sample efficiency and asymptotic performance, demonstrating its remarkable adaptability in complex environments characterized by non-stationary and stochastically evolving tasks. Source code is available at https://github.com/MinWangcs/WISDOM

## 1 INTRODUCTION

Humans excel at dynamically adjusting their behaviors to adapt to continually changing environments over extended periods. A promising paradigm to realize this adaptability in artificial agents is Non-Stationary Reinforcement Learning (NSRL), which focuses on training policies that can adapt sequentially to multiple tasks with varying state transition dynamics or reward functions. Meta-RL enhances sample efficiency and policy generality by identifying shared structures across diverse tasks, making it particularly effective for rapid adaptation. Based on meta-RL, recent NSRL methods have improved task inference through Gaussian mixture distributions, which emphasize the independence of different tasks Bing et al. (2023b) or distance metric losses that enforce stringent smoothness conditions Sodhani et al. (2022). However, they generally perform poorly on non-stationary tasks due to meta-RL's inherent stationarity assumption and its neglect of temporal correlations between tasks.

In NSRL, the task evolution process[1] is typically modeled as a history-dependent stochastic process, implying that task changes follow certain trends or periodic patterns, thus allowing the tracking and prediction of task evolution Xie et al. (2021); Poiani et al. (2021); Chen et al. (2022). A line of work Xie et al. (2021); Ren et al. (2022) attempts to explicitly model the task evolution process as a first-order Markov chain, which limits its applicability to smoothly evolving non-stationary

---

*Corresponding author.
[1] Time-evolving changes of tasks in dynamics or rewards that result in non-stationarity.

tasks, making it ineffective for more complex tasks with rapid changes, where it can result in the accumulation of prediction errors. Bing et al. (2023a) employs recurrent-based neural networks to handle non-stationary sequential data. Poiani et al. (2021); Chen et al. (2022); Tennenholtz et al. (2023) assume a history-dependent task evolution process and approximate it with Gaussian Process (GP) Seeger (2004) or planning in a latent space. The non-stationary kernel function adapts GP to varying covariance, but requires prior knowledge and introduces more parameters. In addition, most existing methods have been primarily limited to task evolution processes that exhibit a regular pattern with fixed periods. However, practical non-stationary tasks generally exhibit time-varying periods or frequencies, posing a challenge for handling such irregular patterns with stochastic periods.

As in Fig. 1 (a), a noisy non-stationary signal changes with increasing frequency, with its three input stages (A, B, C) having similar means ($\approx 0$) and variances ($\approx 2$). Although indistinguishable in the time domain, they can be distinguished in the Fourier spectrum with different main frequencies in Fig. 1 (b). However, the Fourier Transform (FT) Cooley & Tukey (1965) does not reveal when each frequency component appears. For instance, reversing the sequence to create a fast-to-slow pattern (C-B-A), thereby generating a different signal, yields the same Fourier spectrum. In contrast, Wavelet Transform (WT) Polikar et al. (1996) can resolve this by simultaneously preserving time-frequency domain information of the non-stationary signal (initial *approximation coefficient*) and iteratively extracting multi-scale features using different frequencies. Based on *sampling theorem* Shannon (1949), each decomposition of the approximation coefficient halves the sequence length, reducing the amount of data without losing fundamental features. High-frequency noise (*detail coefficient*) in Fig. 1 (c-d) is gradually separated from low-frequency components (*approximation coefficient*), and after the second decomposition in Fig. 1, the smooth low-frequency feature (Fig. 1 (e)) basically reflects the original evolving trend (Fig. 1 (a)).

Inspired by the success of WT theory in handling non-stationary time series by gradually separating different frequency trends Polikar et al. (1996); Yang et al. (2024), we propose **W**avelet-**I**nduced task repre**S**entation pre**D**iction for n**O**n-stationary reinforce**M**ent learning (WISDOM), which leverages wavelet domain information from the task representation sequence **z** to extract underlying structural and temporal patterns. Specifically, we start by encoding trajectories $(s_i, a_i, s_{i+1}, r_i)_{i=0}^{T}$ to obtain **z** to imply the changes of tasks Xie et al. (2021); Chen et al. (2022). Each $z_i$ in this **z** sequence preserves the influence of dynamics and rewards. Each dimension of $z_i$ corresponds to a variate in a multivariate time series. Subsequently, we utilize a wavelet representation network to transform **z** into the wavelet domain. Iterative decompositions of **z** yield an approximation coefficient that captures slowly changing features and a series of detail coefficients that capture rapidly evolving local features. To remove noise and preserve more rapid task changes, we selectively filter detail coefficients through downsampling. We then transform these wavelet coefficients back to the time domain to restore more intrinsic task representations. Moreover, we design a wavelet operator for

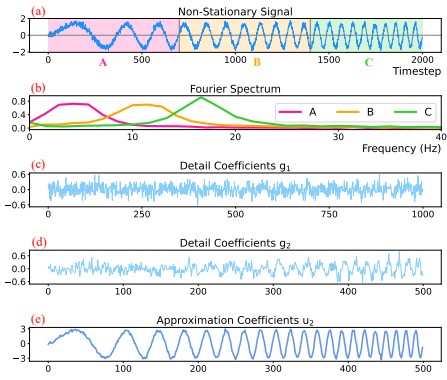

Figure 1: A motivating example.

the explicit temporal difference (TD) update of the wavelet representation network to help capture changes in task structures. Additionally, an auto-regressive loss is incorporated to reinforce long-term temporal dependencies, empowering task evolution predictions. Finally, we integrate restored task representations into policy learning, facilitating prompt policy adjustments and improving policy adaptability across diverse non-stationary tasks. Our key contributions include:

1. We first propose to address non-stationary RL via perceiving its evolution process in the wavelet domain. By iteratively performing wavelet decomposition, we can capture different evolutionary patterns emerging with stochastic period by utilizing a series of varying resolutions or frequencies. This allows WISDOM to dynamically adjust its behavior in response to evolving task conditions, leading to rapid adaptability and improved policy performance.

2. We theoretically demonstrate that wavelet-domain features can serve as informative indicators of policy performance. And our wavelet task representations are demonstrated to lead to improved policies. To better capture structural regularities of the task evolution process, we design a wavelet TD update operator, theoretically ensuring its convergence.

3. We perform a comprehensive evaluation on broader task-distributed Meta-World Yu et al. (2020), Type-1 Diabetes Xie (2018), and MuJoCo Todorov et al. (2012), and the overall results demonstrate the superior and efficient adaptability of WISDOM compared to baselines. We provide new tasks on these benchmarks and open-source for evaluation in challenging non-stationary settings.

## 2 RELATED WORK

**Wavelet Features-Based DL.** Recent studies primarily investigate the incorporation of wavelet features for image processing Yu et al. (2021); Yao et al. (2022), audio signal analysis Pan et al. (2022); Shi et al. (2023); Zhang et al. (2024b), and time series prediction Zhou et al. (2022); Minhao et al. (2021). In addition, some studies Bolós et al. (2020); Kong et al. (2022) have turned to wavelet transforms to integrate time-frequency features for handling non-stationary signals. A wavelet neural network Stock & Anderson (2022) has been proposed to learn a specialized filter-bank for non-stationarity modeling. Yang et al. 2024 designs a graph spectral wavelet to capture the high-frequency components of non-stationary graph signals. To address the issue of online label shift, Qian et al. 2024 proposes a streaming wavelet operator that estimates environmental changes for efficient non-stationary learning. Furthermore, a wavelet attention network is introduced to analyze heterogeneous time series Wang et al. (2023) or even predict seasonal components Wan et al. (2024).

**Non-Stationary RL.** A significant strand of recent research frames the problem of non-stationarity within the meta-RL framework. Al-Shedivat et al. 2018 makes an initial endeavor to apply gradient-based meta-RL to address non-stationary tasks. Meta-experience replay Riemer et al. (2019) and parameter learning rate modulation Gupta et al. (2020) are introduced to enhance adaptability. However, policy gradient updates are typically sample inefficient, and they primarily focus on combining context-based meta-RL to achieve faster adaptation in non-stationary tasks. To improve task representation, Gaussian Mixture Models Bing et al. (2023b), task-metric approaches Sodhani et al. (2022), causal graph Zhang et al. (2024a); Feng et al. (2022), and recurrent neural networks Poiani et al. (2021) have been successively leveraged to assist the context encoder in extracting invariant features. To accurately model the task evolution process, some approaches Xie et al. (2021); Ren et al. (2022) hypothesize the latent context sequence to follow a Markov chain formulation. While such methods demonstrate superior adaptability in smoothly changing tasks, they struggle to adapt effectively to complex tasks that involve sudden changes. Therefore, another branch of research Poiani et al. (2021); Tennenholtz et al. (2023); Chen et al. (2022) assumes history-dependence of the latent context and seeks to utilize Gaussian Processes Seeger (2004), the Rescorla-Wagner model Rescorla (1972) and reward functions Chen et al. (2022) to better characterize the task evolution process. Change-point detection methods represent another line of research for dealing with non-stationary tasks. They utilize sets of partial models specialized for different environments Da Silva et al. (2006), ensembles of context-dynamics predictors that capture different modes of task distributions Alegre et al. (2021), or Wasserstein distances between behavior trajectories Liu et al. (2024) to detect when the task has changed. Different from prior methods, we introduce a principled mechanism that tracks and adapts to task dynamics directly in the wavelet domain, rather than relying solely on implicit learning from data. Through iterative wavelet decomposition, our method naturally separates evolving patterns across multiple frequencies, faithfully capturing the underlying structure of non-stationary tasks.

## 3 PRELIMINARY

### 3.1 CONTEXT-BASED META-RL

Meta-RL is defined on a distribution of stationary tasks $p(\mathcal{T})$, where each task is an MDP represented by a tuple $(\mathcal{S}, \mathcal{A}, \mathcal{P}, \mathcal{R}, \gamma, \mathcal{P}(s_0))$, in which $\mathcal{S}$ denotes the state space, $\mathcal{A}$ the action space, $\mathcal{P}$ the transition dynamics, $\mathcal{R}$ the reward function, $\mathcal{P}(s_0)$ the initial state distribution, and $\gamma \in [0, 1)$ the discount factor Sutton & Barto (1998). In meta-RL, both $\mathcal{P}$ and $\mathcal{R}$ are assumed to be unknown, with different tasks differing only in $\mathcal{P}$ or $\mathcal{R}$. In context-based meta-RL, the task representation $z$ is derived from past transition histories (referred to as context) by a context encoder $e_\eta$. Building on task inference, recent NSRL methods focus on modeling the evolution of $z$ to improve the accuracy of trend predictions, thereby enhancing the adaptability of the policy $\pi(a|s, z)$. In a conventional meta-RL setting, the agent can interact with numerous MDPs, yet the $\mathcal{P}$ and $\mathcal{R}$ of these MDPs remain

constant over time, indicating stationary MDPs. In contrast, NSRL requires the agent to engage with multiple MDPs where $\mathcal{P}$ and/or $\mathcal{R}$ evolve over time, introducing new challenges for rapid adaptation.

## 3.2 Non-stationary RL

Non-stationary RL is defined on the distribution of time-evolving tasks $p(\mathcal{T}_t)$, where an agent interacts with a sequence of MDPs $\mathcal{M}_{\omega_0}, \mathcal{M}_{\omega_1}, \cdots, \mathcal{M}_{\omega_h}$. The evolution of these MDPs is determined by a history-dependent stochastic process $\rho$, i.e., $\omega_{h+1} \sim \rho(\omega_{h+1}|\omega_0, \omega_1, \ldots, \omega_h)$, where $\omega$ is a task ID that regulates the properties of different MDPs Poiani et al. (2021). To better adapt to more practical scenarios, $\omega$ no longer undergoes inter-episode changes, nor does it follow equidistant intra-episode changes. Instead, in our setting, and consistent with Bing et al. (2023b); Chen et al. (2022), $\omega$ varies within episodes with a relatively stochastic period. The overall objective of NSRL is formulated as:

$$\underset{\pi}{\arg\max} \mathbb{E}_{\omega_h} \left[ \sum_{t=0}^{\infty} \mathbb{E}_{\mathcal{T}_t} \left[ \sum_{h=0}^{\infty} \gamma^t r_t^h \mid \mathcal{M}_{\omega_h}, \pi \right] \right]. \tag{1}$$

For example, consider a Walker task in MuJoCo where the agent's mass changes twice during interaction, resulting in three distinct transition dynamics corresponding to MDPs $\omega_0$, $\omega_1$, and $\omega_2$. Their durations are $T_0$, $T_1$, and $T_2$, with $h \in \{0, 1, 2\}$ and timestep $t$ ranging from 0 to $T_0+T_1+T_2-1$. The goal of non-stationary RL is to find a policy that maximizes the expected cumulative reward over this entire trajectory, with a horizon of $T_0 + T_1 + T_2$.

## 3.3 Wavelet Transform

The Wavelet Transform (WT) Burrus et al. (1998) analyzes the various frequency components of a non-stationary signal utilizing a scalable and translatable mother wavelet function $\psi(t)$. To achieve multi-resolution analysis, the Discrete Wavelet Transform (DWT) Shensa et al. (1992) is introduced to decompose the signal layer by layer at different resolutions. Let the initial approximation coefficient $\mathbf{u}_0$ be equal to an input discrete time series $x(n)$. By substituting a pair of low-pass filter $y_0$ and high-pass filter $y_1$ for $\psi(t)$, the following iterative form is derived through **discrete convolution**:

$$\mathbf{u}_m(n) = \sum_{k=1}^{K} y_0(k)\mathbf{u}_{m-1}(2n-k), \quad \mathbf{g}_m(n) = \sum_{k=1}^{K} y_1(k)\mathbf{u}_{m-1}(2n-k), \tag{2}$$

where $m$ is the decomposition level and $k$ is the resolution size. Approximation coefficients $\mathbf{u}_m$ represent low-frequency components, whereas detail coefficients $\mathbf{g}_m$ represent high-frequency components. These multi-resolution and localization properties allow the DWT to simultaneously capture both the global features and local variations of the signal, making it well-suited for handling non-stationary signals, where frequency and period change over time. We also provide a derivation of DWT in Appendix A.1 to enhance understanding.

## 4 Methodology

Sec. 4.1 first outlines the setting of non-stationary tasks and describes how to conduct task inference. Subsequently, Sec. 4.2 elaborates on the implementation of a trainable wavelet representation network that converts task representation sequences into the wavelet domain to capture inherent evolving patterns. Thereafter, Sec. 4.3 devises a wavelet temporal difference (TD) update operator to collaborate with the auto-regressive (AR) loss in optimizing the wavelet representation network. Finally, Sec. 4.4 explains how to effectively harness this inherent structural representation in policy learning.

### 4.1 Module A: Non-Stationary Task Inference with Context Encoder

In scenarios with non-stationary task distributions $p(\mathcal{T}_t)$, assuming the agent has collectively undergone $H$ instances/times of MDP evolution when interacting with the environment, the resulting trajectory, composed of a recent history of transitions $c_{0:T}^{0:H}$ (abbreviated as context $\mathcal{C}$), corresponding to $H + 1$ MDP segments (i.e. $\mathcal{M}_{\omega_0}, \mathcal{M}_{\omega_1}, \ldots, \mathcal{M}_{\omega_H}$). To further enhance applicability, in our non-stationary setup, the period of each $\mathcal{M}_{\omega_h}$, denoted as $T_h$, is assumed unknown. The current

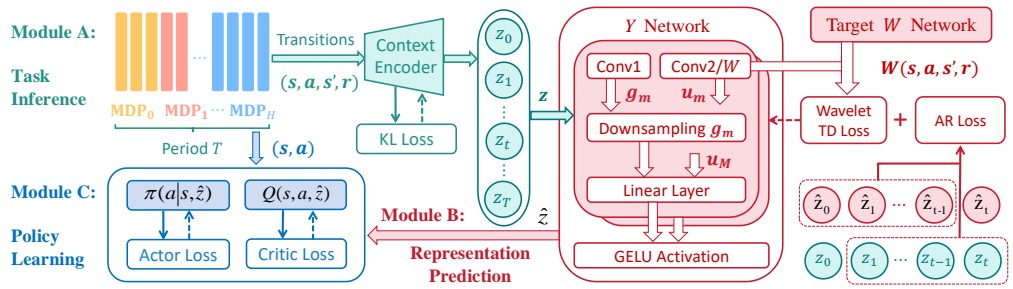

Figure 2: **The architecture of WISDOM.** Module A begins with task inference to derive time-domain task representation $z$. Then module B transforms $z$ into wavelet domain by a wavelet representation network $Y_\phi$, jointly optimized by wavelet TD loss and AR loss to derive wavelet task representation $\hat{z}$. Finally, module C integrates $\hat{z}$ to adjust the policy based on predicted evolving trend.

MDP duration $T_h$ potentially differs from its predecessors $T_{h-1}$, and $T = \sum_{h=0}^{H} T_h$. Each transition $c = (s, a, s', r)$ within $\mathcal{C}$ encapsulates the current state $s$, action $a$, next state $s'$, and the reward $r$. [2]

To enhance practicality, we assume that the agent lacks prior knowledge of the environment and that the true task ID $\omega$ is inaccessible. We employ a context encoder $e$ parameterized by $\eta$ to infer task-relevant information. The encoder $e_\eta$ processes $\mathcal{C}$ to generate a **task representation sequence** $\mathbf{z} = [z_0, z_1, \ldots, z_T]$ that is generally viewed as an approximation of the $\omega$ sequence Xie et al. (2021); Chen et al. (2022), where each $z$ uniquely corresponds to each transition $c$. The KL-divergence is used as a variational approximation to an information bottleneck Rakelly et al. (2019) to train $e_\eta$:

$$\mathcal{J}_\eta = \mathbb{E}_{\mathcal{C} \sim \mathcal{B}}[D_{\text{KL}}(e_\eta(\mathbf{z}|\mathcal{C}) \| p(\mathbf{z}))], \tag{3}$$

where $p(\mathbf{z})$ represents a Gaussian prior, and the context $\mathcal{C}$ is sampled from the replay buffer $\mathcal{B}$.

### 4.2 MODULE B: TRACKING TASK EVOLUTION VIA WAVELET REPRESENTATION

In NSRL, the task sequence $\mathcal{M}_{\omega_0 : \omega_H}$ is generally modeled as a history-dependent stochastic process Xie et al. (2021); Chen et al. (2022); Poiani et al. (2021). From a time series analysis perspective, this sequence can be viewed as a non-stationary signal, with the underlying frequency of each type of task $f_h$ changing over time. WT is widely recognized for its ability to process non-stationary signals by decomposing them into different frequency components (multi-scale features) Polikar et al. (1996). Conceptually supported by Xie et al. (2021); Poiani et al. (2021), WISDOM is the first to address NSRL by leveraging wavelet-domain task representations to track the temporal evolution of tasks.

Inspired by the design of prevalent WT networks Oord (2016); Shi et al. (2023), our **wavelet representation network** $Y_\phi$ consists of two dilated causal convolution networks, ***Conv1*** and ***Conv2***, followed by a linear layer to guarantee the capture of underlying task dynamics. Specifically, $Y_\phi$ performs the DWT in Eq. 2 with recursive convolution and transforms $\mathbf{z}$ into the wavelet domain:

$$\mathbf{g}_m = \textbf{\textit{Conv1}}(\mathbf{u}_{m-1}, y_1; M), \quad \mathbf{u}_m = \textbf{\textit{Conv2}}(\mathbf{u}_{m-1}, y_0; M), m \in [1, M], M \in \mathbb{N}^+, \mathbf{u}_0 = \mathbf{z}, \tag{4}$$

where the learnable convolution kernels $y_0$ and $y_1$ can be initialized as classical basis functions such as Haar wavelet[3]. Hence, $Y_\phi$ is guided to approximate traditional wavelet behavior while adaptively adjusting the kernel values during training. $y_0$ functions as a low-pass filter, eliminating high-frequency components, thereby enabling the approximation coefficient $\mathbf{u}_m$ to capture the overall trends of non-stationary task evolution. Conversely, $y_1$ acts as a high-pass filter, gathering high-frequency components, allowing the detail coefficients $\mathbf{g}_m$ to capture rich detail features of tasks. After performing $M$ decompositions on $\mathbf{u}_m$, $Y_\phi$ obtains the $M$th $\mathbf{u}_M$, along with the total $M$ detail coefficients $\mathbf{g}_{1:M}$. During each decomposition step $m$, the network downsamples and preserves the most recent detail coefficients $\tilde{\mathbf{g}}_m$. Finally, $Y_\phi$ applies a linear transformation to revert $\tilde{\mathbf{g}}_m$ and $\mathbf{u}_M$ to the time domain, yielding a more expressive and intrinsic **wavelet task representation sequence** $\hat{\mathbf{z}}$.

---

[2]Since the time steps of changes in the MDP is unknown, superscript $h$ is omitted in the following notation.

[3]For haar wavelet Pattanaik & Bouatouch (1995), $y_0 = [1/\sqrt{2}, 1/\sqrt{2}]$ computes the average over adjacent elements, smoothing the $\mathbf{z}$ sequence, akin to a low-pass filter. In contrast, $y_1 = [1/\sqrt{2}, -1/\sqrt{2}]$ computes the difference, highlighting local changes and extracting high-frequency components, similar to a high-pass filter.

### 4.3 Optimization Objective Design for Wavelet Representation Network $Y_\phi$

We propose a TD-style optimization of the representation in the wavelet domain and theoretically demonstrate the convergence of the corresponding wavelet TD operator, thereby facilitating learning and search for task-oriented representations.

In wavelet theory, the ***Conv2*** layer (referred to as the $W$ network hereafter) integrates a low-pass filter to learn stable and inherent non-stationary features Shensa (1992). We define a TD-style update operator over wavelet-based task features learned by the $W$ network as $\mathcal{F}W(\mathbf{z}_t) = \mathbf{z}_t + \Gamma\mathbb{E}_\pi[W(\mathbf{z}_{t+1})]$ to explicitly update $Y_\phi$, where $\Gamma$ denotes the discount factor in diagonal matrix form. Similar to successor features Barreto et al. (2017), wavelet features $W(\mathbf{z}_t)$ satisfy the Bellman equation. Theorem 1 demonstrates that $\mathcal{F}$ is a contraction mapping (Proof in Appendix A.2), ensuring the convergence of wavelet representation updates and allowing for stable and consistent learning. Unlike most existing work that implicitly updates the representation network with TD loss over the value function, our explicit TD update will not neglect low-reward yet critical features. Although rewards in many RL settings can be sparse or delayed, learned representations tend to be denser and more informative. Intuitively, the wavelet TD update allows $Y_\phi$ to better capture structural regularities in MDP sequences and enhance sample efficiency.

Moreover, the wavelet TD loss relies solely on the single-step future task representation $z_{t+1}$ and provides a more concise optimization objective than the commonly used auto-regressive (AR) loss in time-series forecasting Shi et al. (2023); Oord (2016). Our $W$ network is also implemented with dilated causal convolution, ensuring that the $i$-th output of $Y_\phi$ only depends on the first $i$ input elements, thus maintaining the conditional dependency essential for AR modeling to predict task changes. However, AR objectives often suffer from accumulated prediction errors, where errors in the current representation propagate and degrade future predictions. As a remedy, the wavelet TD update with the target $W$ network helps mitigate error propagation and stabilizes optimization through delayed target update.

**Theorem 1** *Let $\mathcal{W}$ denote the set of all functions $W : \mathcal{S} \times \mathcal{A} \to \mathbb{C}^D$ that map from the time domain to the wavelet domain. The wavelet update operator $\mathcal{F} : \mathcal{W} \to \mathcal{W}$, defined as*

$$\mathcal{F}W(\mathbf{z}_t) = \mathbf{z}_t + \Gamma W(\mathbf{z}_{t+1}),$$

*is a contraction mapping, where $\mathbf{z}_t$ and $\mathbf{z}_{t+1}$ represents the current and next task representation sequence, respectively, and $\Gamma$ denotes the discount factor in a diagonal matrix form.*

The overall optimization objective of wavelet representation network $Y_\phi$ is formulated as follows:

$$\mathcal{J}_\phi = \alpha_Y \underbrace{\mathbb{E}_{c \sim \mathcal{B}}\left[\frac{1}{2}\left(W_\varphi(\mathbf{z}_t) - (\mathbf{z}_t + \Gamma\mathbb{E}_\pi[W_{\bar{\mu}}(\mathbf{z}_{t+1})])\right)^2\right]}_{\text{Wavelet TD loss}} - \underbrace{\mathbb{E}_{\hat{z} \sim Y_\phi}\left[\log\prod_{t=0}^{T} P(\hat{z}_t | \hat{z}_{<t})\right]}_{\text{AR loss}}, \quad (5)$$

where $\alpha_Y$ balances the contribution of each loss. Similarly to DQN Mnih et al. (2013), $W_{\bar{\mu}}$ is a target network to stabilize the training and is updated separately via an exponential moving average of the parameters of the $W_\varphi$ network. Complemented to the wavelet TD loss, the AR loss adheres to stricter temporal constraints, preventing the trends captured by $Y_\phi$ from being temporally misaligned due to the relaxed orthogonality of the learnable filters. Moreover, it reinforces long-term temporal dependencies, empowering $Y_\phi$ to make predictions based on the task representation sequence. Together, they facilitate rapid adaptation to evolving task structures.

### 4.4 Module C: Policy Learning Through Wavelet Task Representations

To establish a more intimate relationship between dynamic adjustment of the policy in advance and the non-stationary evolving trends, we further incorporate the predicted wavelet task representation $\hat{z}$ generated by $Y_\phi$ into the policy iteration. Our WISDOM is based on the Soft Actor-Critic (SAC) algorithm Haarnoja et al. (2018) and can also be integrated with any downstream RL algorithms.

The objective of training the contextual critic network $Q_\upsilon$ is to minimize the squared residual error:

$$\mathcal{J}_\upsilon = \mathbb{E}_{(s,a) \sim \mathcal{B}, \hat{z} \sim Y_\phi}\left[\frac{1}{2}\left(Q_\upsilon(s, a, \hat{z}) - Q_{\text{target}}\right)^2\right], \quad (6)$$

with the target $Q$ value defined as $Q_{\text{target}} = r + \gamma \mathbb{E}_{s' \sim \mathcal{B}, a' \sim \pi_\theta, \hat{z} \sim Y_\phi} \left[ Q_{\bar{\zeta}} \left( s', a', \hat{z} \right) \right]$, where $\bar{\zeta}$ denotes stopping the backpropagation of gradients of the target critic network $Q_\zeta$. The parameters of $Q_\zeta$ are updated with an exponential moving average derived from the weights of the $Q_\upsilon$ network. The contextual policy network $\pi_\theta$ can be updated by optimizing the following objective:

$$\mathcal{J}_\theta = \mathbb{E}_{s \sim \mathcal{B}, a \sim \pi_\theta, \hat{z} \sim Y_\phi} \left[ \alpha \log \left( \pi_\theta \left( a|s, \hat{z} \right) \right) - Q_\upsilon \left( s, a, \hat{z} \right) \right], \qquad (7)$$

where $\alpha$ is a temperature coefficient. Alg. 1 in Appendix C presents the pseudocode of WISDOM. The following theorems establish a connection between task representations through the wavelet transform and policy performance. In Theorem 2, we demonstrate that the wavelet-domain task representations preserve the ability to indicate policy performance. And in Theorem 3, we prove that the policy is improved with restored time-domain task representations during policy iteration.

**Theorem 2** *Suppose that the reward function $\mathcal{R}(z)$ can be expanded into a Bth-degree Taylor series for $z \in \mathbb{R}^D$, then for any two policies $\pi_1$ and $\pi_2$, their performance difference is bounded as:*

$$|J_{\pi_1} - J_{\pi_2}| \leq \frac{\sqrt{D}}{1 - \gamma} \cdot \sum_{b=1}^{B} \frac{\left\| \mathcal{R}^{(b)}(0) \right\|_D}{b!} \cdot \max_{1 \leq q \leq D} \sup_{d_q \in \mathbb{R}, \beta_q > 0} \left| \mathcal{W}_{\pi_1}^{(b)}(d_q, \beta_q) - \mathcal{W}_{\pi_2}^{(b)}(d_q, \beta_q) \right|, \quad (8)$$

*where $\mathcal{W}_\pi^{(b)}(d, \beta)$ denotes the wavelet transform of the bth power of the task representation sequence $\mathbf{z}^{(b)} = [z_0, z_1, \ldots, z_{T_H}]^{(b)}$ for any integer $b \in [1, B]$, and $J_\pi$ represents the average policy performance over the entire task space within a single episode.*

See Appendix A.3 for the proof. Theorem 2 describes how the performance differences of the wavelet-domain features control the performance of the corresponding policies. We prove that wavelet-domain features can serve as an informative indicator of policy performance, aiming at more efficient policy iteration to find the optimal policy and exhibiting fast convergence in experiments.

**Theorem 3** *WISDOM returns a policy $\pi(\cdot|s, \hat{z})$ conditioned on the wavelet task representation that improves upon its intermediate policy produced during policy iteration $\pi_h$, satisfying $J_{WISDOM} \geq J_{\pi_h}$ under certain assumptions, where $J_{WISDOM}$ denotes the final policy performance of WISDOM.*

The assumptions and proof are provided in Appendix A.4. Theorem 3 describes that our restored $\hat{z}$ in the time domain after wavelet transform leads to an improved contextual policy during iteration. The wavelet transform is well known for enhancing the signal-to-noise ratio (SNR) by separating different frequencies Polikar et al. (1996). And $\hat{z}$ is expected to filter out task-irrelevant information and helps the policy focus on essential non-stationary characteristics, providing a clearer optimization direction.

## 5 EXPERIMENTS

**Task settings.** Following SeCBAD Chen et al. (2022), the stochastic period $T_h$ of an MDP $\mathcal{M}_{\omega_h}$ is sampled from a Gaussian distribution with a mean of 60 and a variance of 20 time steps. We mainly performed experiments on the following three benchmarks, and more details are in Appendix B.

1. **Meta-World** Yu et al. (2020), consisting of 50 robotic manipulation tasks with a broader distribution that is generally considered challenging, exhibits non-stationarity in the continuous variation of the target position for the robotic arm movement, which is correlated with the reward function;

2. **Type-1 Diabetes** Xie (2018) aims to test the control of blood glucose levels (observation) in type-1 diabetic patients by injecting insulin (action). The non-stationarity is reflected in the continuous variation of food intake, which affects blood glucose levels and the dynamics of tasks;

3. **MuJoCo** Todorov et al. (2012), widely adopted in NSRL, has only parametric diversity. We considered two MuJoCo scenarios: a) Evolution tasks with changing reward functions (e.g., Walker-Vel); b) Evolution tasks with changing dynamics (e.g., Cheetah-Damping).

**Baselines.** We compared WISDOM with four competitive NSRL baselines: CEMRL Bing et al. (2023a), TRIO Poiani et al. (2021), SeCBAD Chen et al. (2022), and COREP Zhang et al. (2024a), which utilize the Gaussian Mixture Model, Gaussian Process, reward function, and causal graph to model the task evolution process, respectively. Additionally, we compared SAC Haarnoja et al.

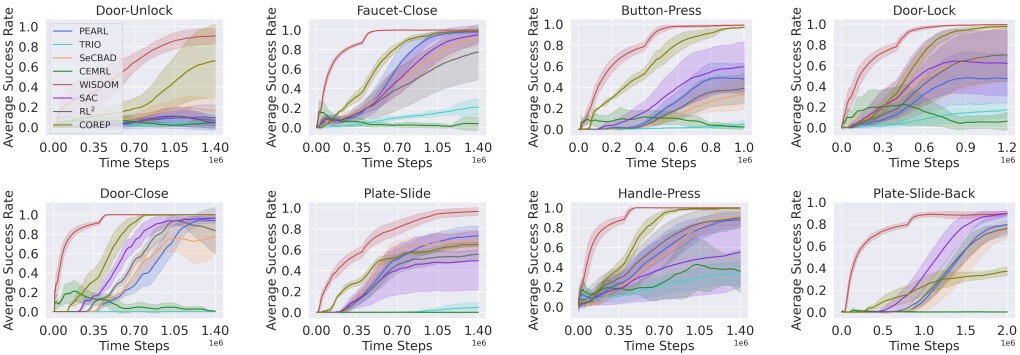

Figure 3: Testing average performance on Meta-World over 6 random seeds. Our WISDOM achieves rapid convergence and exhibits excellent asymptotic performance.

(2018), PEARL Rakelly et al. (2019) and RL$^2$ Duan et al. (2016) to highlight the importance of task inference and modeling of the task evolution process, respectively.

**Meta-World results.** Fig. 3 and Table 1 illustrate the testing curves and converged performance, respectively. As in Fig. 3, WISDOM consistently demonstrates exceptional adaptation efficiency across all environments, surpassing all baseline methods. As in Table 1, WISDOM achieves the highest average success rate with minimal variance in most environments, significantly contributing to effective noise filtering and the retention of fundamental low-frequency components. Although SeCBAD outperforms PEARL in Door-Unlock and Door-Lock, it falls behind PEARL regarding rapid adaptability in most environments. In Meta-World, where the evolutionary factors are relatively complex due to the potential variations in the three-dimensional coordinates of the target position, agents require substantial trial and error during the early training due to fluctuating rewards. Consequently, SeCBAD faces challenges in relying solely on rewards to identify whether task evolution has occurred. Surprisingly, PEARL and RL$^2$ have impressive overall converged performance, indicating their potential to adapt to non-stationary tasks by solely meta-learning a policy, even without explicitly modeling the inherent changing characteristics of task evolution. The competitive SAC further illustrates that inaccuracies in the inference and modeling of non-stationary tasks in more challenging environments may undermine the adaptability of the policy. When CEMRL attempts to cluster complex task distributions into a few categories, representation collapse often occurs. Conversely, clustering into a larger number of categories significantly increases the complexity. The selection of kernel functions in TRIO has a similar dilemma, causing wholly poor performance.

Table 1: Converged average test success rate $\pm$ standard error (%) on Meta-World.

|  | Door-Unlock | Faucet-Close | Button-Press | Door-Lock | Door-Close | Plate-Slide | Handle-Press | Plate-Slide-Back |
|---|---|---|---|---|---|---|---|---|
| CEMRL | 4.08±8.74 | 0.17±0.37 | 1.83±4.10 | 6.67±14.14 | 0.42±0.93 | 0.00±0.00 | 45.17±38.58 | 0.00±0.00 |
| TRIO | 3.92±6.46 | 20.17±22.14 | 10.42±16.36 | 20.75±19.64 | 0.00±0.00 | 6.42±10.39 | 33.25±26.92 | 0.20±**0.28** |
| PEARL | 10.25±19.31 | 96.33±6.88 | 39.42±32.54 | 50.67±32.74 | 88.58±14.64 | 73.50±17.18 | 85.33±20.78 | 82.50±13.73 |
| SeCBAD | 11.58±17.01 | 93.42±8.99 | 36.58±32.46 | 72.50±31.95 | 89.33±18.20 | 71.50±19.93 | 93.92±12.37 | 79.53±11.38 |
| COREP | 67.50±45.96 | 98.00±2.21 | 96.83±4.24 | 98.67±1.89 | **100.00±0.00** | 64.17±6.86 | **100.00±0.00** | 43.00±10.19 |
| SAC | 1.67±**3.14** | 98.33±3.73 | 62.83±38.22 | 62.67±39.33 | **100.00±0.00** | 50.00±37.15 | 61.25±45.63 | 90.03±5.98 |
| RL$^2$ | 5.75±9.38 | 81.00±36.88 | 39.83±28.49 | 70.67±35.14 | 81.83±36.92 | 56.92±15.71 | 94.75±9.65 | 79.23±9.85 |
| **WISDOM** | **91.58**±9.36 | **99.92**±0.19 | **99.42**±**1.30** | **99.67**±0.75 | **100.00±0.00** | **96.50**±6.04 | **100.00±0.00** | **90.57**±7.66 |

**Type-1 Diabete results.** We further examined the adaptability of all models in a more realistic environment that investigates the relationship between the control of blood glucose levels and insulin injections in diabetic patients. As observed in Fig. 4, WISDOM demonstrates notably more efficient adaptability, indicating that wavelet task representations can effectively capture the trend of changing blood glucose levels

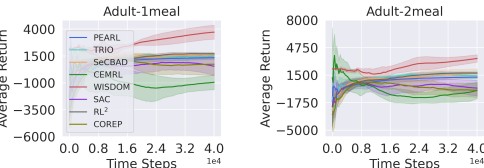

Figure 4: Testing average return on glucose control environments over 6 random seeds.

and aid in selecting appropriate insulin dosages to maintain normal levels. Moreover, SeCBAD and CEMRL struggle to adapt to environments with changing food intake, such as environments where both lunch and dinner (2 meal) quantities vary over time. This difficulty may stem from the limited distinctiveness of the trajectory data, making it challenging to rely on rewards to determine the evolution moment or to cluster the trajectories into limited categories.

**MuJoCo results.** Following CEMRL, the non-stationarity in MuJoCo is simulated by dynamically adjusting parameters, such as target velocity. However, these adjustments are relatively small, leading to a narrower distribution of non-stationary tasks. Consequently, these tasks tend to be less challenging compared to those in Meta-World. Besides, the lower state dimension

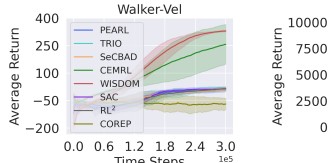

Figure 5: Testing return on MuJoCo over 6 seeds.

of MuJoCo results in relatively simple relationships between the graph nodes, which explains why COREP's advantage over other baselines is not as significant as in Meta-World. In Fig. 5, CEMRL demonstrates enhanced capability in clustering the task distribution, which facilitates precise classification of data in the replay buffer and effective extraction of task-specific representations. Since both TRIO and SeCBAD explicitly model the task evolution process, and TRIO additionally incorporates a recurrent encoder to capture and leverage history information, they converge faster than PEARL and $RL^2$. Significantly, WISDOM shows consistently rapid adaptability and enhanced final performance.

**Ablation studies.** We conduct ablation studies to validate the role of the wavelet representation network ($Y$ network) and its optimization objective, and the structure of the context encoder. In Fig. 6(a), the $Y$ network provides significant gains, further indicating that the wavelet task representation reflects non-stationary trends. The AR loss accelerates convergence and improves final performance, and the wavelet TD loss stabilizes training and reduces variance. In contrast, the RNN encoder tends to forget changes over time and is susceptible to gradient vanishing, whereas the MLP encoder (WISDOM) shows greater stability. Since each $z$ in $\mathbf{z}$ is generally multidimensional, each dimension of $z$ can be viewed as a variate in the multivariate time series. In addition to performing feature concatenation at each time step when performing DWT as WISDOM, we investigate a *Variable-Wise Encoding (VWE)* strategy which applies DWT to the sequences constructed from each variate independently. The *VWE* strategy leads our model to converge more slowly and yields inferior final results. We attribute this to the loss of cross-variate interaction modeling caused by disrupting the dependencies among variates. In the time series domain, this *VWE* strategy often necessitates additional structural components to re-establish inter-variate relationships.

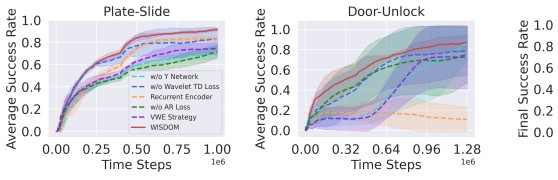 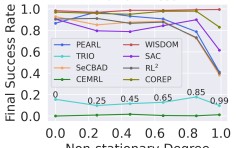 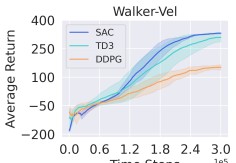

(a) Ablation study of WISDOM.  (b) Impact of N-S degrees. (c) Impact of different RL.

Figure 6: Ablation study and evaluation of different non-stationary (N-S) degrees and RL backbones.

**Adaptability analysis of different non-stationary degrees and RL backbones.** The non-stationary degrees[4] we set in Meta-World, MuJoCo, and Type-1 Diabetes are 0.99, 0.97, and 0.7, respectively. In Fig. 6(b), most models' final performance shows a downward trend as non-stationarity increases. Contrastively, our WISDOM demonstrates remarkable and consistent adaptability, highlighting the more expressive wavelet task representation and robust policy. In Fig. 6(c), WISDOM can quickly adapt and converge utilizing various RL algorithms. Due to limited exploration, DDPG often converges to a local optimum. For fair comparisons, all models employ SAC as the backbone.

**Robustness to noise for WISDOM and baseline methods.** We conducted experiments by injecting Gaussian noise $N(0, 1)$ into states to evaluate robustness. As illustrated in Fig. 7, all baselines exhibit slower convergence and reduced performance under noisy conditions. In contrast, our WISDOM maintains the highest success rate and rapid convergence. We credit this to the wavelet-based representation learning process, which not only suppresses noise but also retains fast-varying and task-relevant signals more effectively, thereby improving the signal-to-noise ratio.

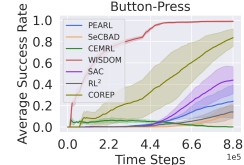

Figure 7: Evaluation of robustness to noise.

---

[4]Measured as $(T - \overline{T_h})/T \in [0, 1)$, where $T$ is the total period and $\overline{T_h}$ denotes the mean of the stochastic period of an MDP $\mathcal{M}_{\omega_h}$. Larger values of degrees indicate severer non-stationarity.

**Case study.** We visualize the changes of representations tracked by the $Y_\phi$ network to validate its ability to capture intrinsic non-stationary changes. As depicted in the yellow lines of Fig. 8, the velocity of the agent in Inverted Double Pendulum undergoes 7 changes over 440 consecutive time steps, generating a non-stationary sequence $\mathcal{M}_{\omega_{0:7}}$ with varying state transition dynamics.

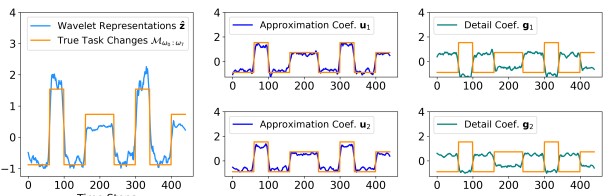

Figure 8: The case study shows that the wavelet task representation $\hat{z}$ accurately reflect the true changes of task $\mathcal{M}_\omega$.

In the middle subfigures, the approximation coefficients $\mathbf{u}$ that represent low-frequency trends match the true underlying task changes and become smoother with deeper decomposition. In the right subfigures, the detail coefficients $\mathbf{g}$ that represent high-frequency details show sharper, more localized changes and may appear inverted due to the orthogonality of the initialized Haar filters. As shown in the left subfigure, when we combine the final approximation $\mathbf{u_2}$ with selected details, we obtain wavelet task representation $\hat{z}$, which closely aligns with the true trend of the non-stationary task. Overall, this illustrates that the wavelet-based representation successfully captures the underlying structure of how tasks change over time.

# 6    CONCLUSION, LIMITATION AND FUTURE WORK

We introduced WISDOM, a novel approach to address the challenge of non-stationarity in RL by leveraging a learnable wavelet representation network to capture the trends of task evolution in the wavelet domain, facilitating flexible adaptation to complex non-stationary tasks with stochastic evolving periods. WISDOM captures more intrinsic evolving features of non-stationary task changes, learns to predict the evolving trend, and ensures efficient and improved policy learning. Experimental results on Meta-World, MuJoCo, and Type-1 Diabetes benchmarks demonstrate its superb adaptability and performance, highlighting its robustness and efficiency in handling non-stationary tasks.

Although WISDOM is effective and easy to implement, it presents certain limitations. The compactness of task representations may be influenced by decomposition levels in the wavelet representation network, necessitating tuning appropriate decomposition levels based on specific environments. Future directions include: 1) Developing adaptive mechanisms for dynamically adjusting decomposition levels; and 2) Exploring effective selection of detail coefficients for rapid adaptation.

## ACKNOWLEDGMENTS

The authors would like to thank the anonymous reviewers for their valuable feedback. This work was partially supported by the National Natural Science Foundation of China (Grant No. 62276024), the Beijing Natural Science Foundation (Grant No. 4262066), the Fundamental Research Funds for the Central Universities, Jilin University (Grant No. 93K172025K01), and the Fundamental Research Funds for the Central Universities (Grant No. 2025CX01010).

## REPRODUCIBILITY STATEMENT

We have taken several steps to ensure that our work is fully reproducible. All experiments are conducted on publicly available benchmarks, with complete descriptions of these benchmarks and their configurations provided in Section 5 and Appendix B. Additional details regarding the tasks and baselines are available in Appendix B. To support reimplementation, we present pseudocode for our methods in Appendix C. The neural network architectures, experimental protocols, and hyperparameters are thoroughly described in Appendix B and Appendix D. For our theoretical results, all proofs are provided in Appendix A. All the code required to train and evaluate the proposed methods has been open-sourced, ensuring full accessibility for the reproduction of our results.

## ETHICS STATEMENT

We place strong emphasis on adhering to ethical standards throughout this research. This study raises no notable ethical concerns, as all experiments were conducted within simulated and well-controlled environments. This research relies exclusively on data generated from simulations, therefore, it does

not raise privacy concerns nor involve experiments posing risks to human subjects. Additionally, we followed strict ethical standards throughout the development of our method, and remain committed to addressing any unintentional errors or lapses in a timely manner.

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

## A    PROOF

### A.1    PROOF OF THE DISCRETE WAVELET TRANSFORM

The Wavelet Transform (WT) Burrus et al. (1998) analyzes and represents the various frequency components of a signal utilizing a scalable and translatable mother wavelet function $\psi$. Given a time series $x(t)$, the continuous wavelet transform (CWT) extracts its frequency by performing the **continuous convolution**:

$$\mathcal{W}(d, \beta) = |\beta|^{-\frac{1}{2}} \int_{-\infty}^{\infty} x(t) \psi \left( \beta^{-1}(t - d) \right) \mathrm{d}t. \tag{9}$$

The translation position $d$ allows the $\psi$ function to move along the signal $x(t)$ in the time domain, enabling analysis at different time points. The scaling factor $\beta$ controls the degree of stretching and compression of $\psi$ function, which in turn affects its resolution properties and the range of extracted frequencies. Specifically, a smaller $\beta$ corresponds to higher resolution and a smaller timescale, facilitating the capture of finer details, while a larger $\beta$ corresponds to lower resolution and a larger timescale, suitable for analyzing overall trends. CWT typically analyzes signals at a fixed resolution, either fine or coarse. To achieve multi-resolution analysis while simplifying computations, the discrete version of CWT, the Discrete Wavelet Transform (DWT) Shensa et al. (1992), is introduced to decompose the signal layer by layer at different resolutions, as formalized in Lemma 1.

**Lemma 1** *DWT. Let the initial approximation coefficient $\mathbf{u}_0$ be equal to an input discrete time series $x(n)$. By substituting a pair of **low-pass filter** $y_0$ and **high-pass filter** $y_1$ for the mother wavelet function $\psi(t)$, following iterative form is derived through discrete convolution operations:*

$$\mathbf{u}_m(n) = \sum_{k=1}^{K} y_0(k)\mathbf{u}_{m-1}(2n-k), \mathbf{g}_m(n) = \sum_{k=1}^{K} y_1(k)\mathbf{u}_{m-1}(2n-k), \quad (10)$$

*where $m$ represents the decomposition number and $k$ denotes the resolution size. The **approximation coefficients** $\mathbf{u}_m$ represent low-frequency components, whereas the **detail coefficients** $\mathbf{g}_m$ represent high-frequency components.*

**Proof 1** *For the continuous time series $x(t), t \in [0, T)$, the coarsest approximation of $x(t)$ is:*

$$x^{(0)}(t) \triangleq \mathbf{u}_{0,0}\psi(t), \quad \psi(t) = 1(0 \le t < T), \quad \mathbf{u}_{0,0} = \int_0^T x(t)dt, \quad (11)$$

*where $\psi(t)$ denotes the scaling function and the coefficient $\mathbf{u}_{0,0}$ is the average value of $x(t)$ in this interval. The superscript 0 implies the initial approximation of $x(t)$. To obtain a more accurate estimation of $x(t)$, the approximation can be refined by halving the intervals into smaller parts:*

$$x^{(1)}(t) \triangleq \mathbf{u}_{1,0}\psi(2t) + \mathbf{u}_{1,1}\psi(2t-1), \quad \mathbf{u}_{1,0} = \int_0^{T/2} x(t)dt, \quad \mathbf{u}_{1,1} = \int_{T/2}^T x(t)dt. \quad (12)$$

*Define the mother wavelet function as:*

$$\psi_{m,k}(t) \triangleq 2^{\frac{m}{2}}\psi\left(2^m t - k\right) \quad (13)$$

*By repeating this procedure, $x(t)$ can be approximated with finer precision, given as:*

$$x^{(m)}(t) = \sum_{k \in \mathbb{Z}} \mathbf{u}_{m,k}\psi_{m,k}(t), \quad \mathbf{u}_{m,k} = \int_{-\infty}^{\infty} x(t)\psi_{m,k}(t)\mathrm{d}t, \quad (14)$$

*where $m \in \mathbb{N}$ represents the $m$-th interval bisection, $k$ represents a series of resolutions, and $\star$ indicates the convolution operation. After the $m$-th bisection of the original time interval, the scaling functions in the subspace $U_m \triangleq \mathrm{span}\left(\{\psi_{m,k}\}_{k\in\mathbb{Z}}\right)$ are capable of capturing local structures in $x(t)$ at a timescale no longer than $T/2^m$.*

*Inspired by multi-resolution analysis (MRA) Willsky (2002), discrete wavelet transform (DWT) Shensa et al. (1992); Shi et al. (2023) introduces the orthogonal subspace $G_m \triangleq \mathrm{span}\left(\{\delta_{m,k}\}_{n\in\mathbb{Z}}\right)$ of $U_m$, where $\delta$ indicates orthogonal scaling functions Burrus et al. (1998). $U_M$ can be decomposed into the orthogonal sum of the lowest resolution subspace $U_0$ and a series of orthogonal complement $G$:*

$$U_M = U_{M-1} \oplus G_{M-1} = U_0 \oplus G_0 \oplus \ldots \oplus G_{M-2} \oplus G_{M-1}. \quad (15)$$

*Accordingly, $x(t)$ can be represented by a series of different resolutions:*

$$x^{(M)}(t) = \mathbf{u}_{0,0}\psi(t) + \sum_{m=0}^{M-1} \sum_{k \in \mathbb{N}} \mathbf{g}_{m,k}\delta_{m,k}(t), \quad (16)$$

*where $\mathbf{u}$ and $\mathbf{g}$ are referred to as the approximation coefficient and detail coefficient, respectively. Therefore, for discrete time series $x(n)$, we can rewrite Eq. 14 in the following iterated form:*

$$
\begin{aligned}
x_{m-1} &= \sum_n \mathbf{u}_{m-1}(n)\psi_{m-1}(n) \\
&= \sum_n \mathbf{u}_{m-2}(n)\psi_{m-2}(n) + \sum_n \mathbf{g}_{m-2}(n)\delta_{m-2}(n),
\end{aligned}
\quad (17)
$$

*where $n$ represents the sequence length, and $\psi(n)$ and $\delta(n)$ are the discrete forms of $\psi(t)$ and $\delta(t)$, respectively. As a result, by performing the discrete convolution operation $(\star)$, the discrete wavelet transform can be obtained as follows:*

$$
\begin{aligned}
\mathbf{u}_m(n) &= x_{m-1} \star \psi_m(n) \\
&= \left(\sum_{n=1}^N \mathbf{u}_{m-1}(n)\psi_{m-1}(n)\right) \star \left(\sum_{k=1}^K y_0(k)\psi_{m-1}(2n-k)\right) \\
&= \sum_{k=1}^K y_0(k)\mathbf{u}_{m-1}(2n-k), \text{ and similarly },
\end{aligned}
\quad (18)
$$

$$\mathbf{g}_m(n) = x_{m-1} \star \delta_m(n) = \sum_{k=1}^K y_1(k)\mathbf{u}_{m-1}(2n-k),$$

*which concludes the proof of lemma 1.*

Non-stationary RL is defined on the time-evolving task distribution $p(\mathcal{T}_t)$, where an agent interacts with a sequence of Markov Decision Processes (MDPs) Sutton & Barto (1998) $\mathcal{M}_{\omega_0}, \mathcal{M}_{\omega_1}, \cdots, \mathcal{M}_{\omega_h}$. The evolution of these MDPs is determined by a history-dependent stochastic process $\rho$, i.e., $\omega_{h+1} \sim \rho(\omega_{h+1}|\omega_0, \omega_1, \ldots, \omega_h)$, where $\omega$ is a task ID that regulates the properties of different MDPs. Each MDP $\mathcal{M}_\omega$ is represented by a tuple $(\mathcal{S}, \mathcal{A}, \mathcal{P}_\omega, \mathcal{R}_\omega, \gamma, \mathcal{P}(s_0))$, in which $\mathcal{S}$ denotes the state space, $\mathcal{A}$ the action space, $\mathcal{P}(s'|s, a)$ the transition dynamics, $\mathcal{R}(s, a, s')$ the reward function, $\mathcal{P}(s_0)$ the initial state distribution, and $\gamma \in [0, 1)$ the discount factor.

## A.2 PROOF OF THE CONVERGENCE OF THE WAVELET TD LOSS

**Theorem 1** *Let $\mathcal{W}$ denote the set of all functions $W : \mathcal{S} \times \mathcal{A} \to \mathbb{C}^D$ that map from the time domain to the wavelet domain. The wavelet update operator $\mathcal{F} : \mathcal{W} \to \mathcal{W}$, defined as*

$$\mathcal{F}W(\mathbf{z}_t) = \mathbf{z}_t + \Gamma\mathbb{E}_\pi[W(\mathbf{z}_{t+1})], \tag{19}$$

*is a contraction mapping, where $\mathbf{z}_t$ and $\mathbf{z}_{t+1}$ represents the current and next task representation sequence, respectively, and $\Gamma$ denotes the discount factor in a diagonal matrix form.*

**Proof 2** *The norm on $\mathcal{W}$ is defined as $\|W\|_{\mathcal{W}} := \sup_{\mathbf{z} \in \mathcal{B}} \max_{\mathbb{K} \in \mathcal{K}} \left\| \left[ W(\mathbf{z_t}) \right]_{\mathbb{K}} \right\|_D$, where $\mathcal{K}$ denotes the sequence length of $\mathbf{z_t}$. For any $W_1, W_2 \in \mathcal{W}$, we have*

$$
\begin{aligned}
\|\mathcal{F}^\pi W_1 - \mathcal{F}^\pi W_2\|_{\mathcal{W}} &= \sup_{\mathbf{z} \in \mathcal{B}} \max_{\mathbb{K} \in \mathcal{K}} \|[\mathbf{z}_t]_{\mathbb{K}} + \gamma\mathbb{E}_{\mathbf{z}_{t+1} \sim \mathcal{B}} [[W_1(\mathbf{z}_{t+1})]_{\mathbb{K}}] \\
&\quad - [\mathbf{z}_t]_{\mathbb{K}} - \gamma\mathbb{E}_{\mathbf{z}_{t+1} \sim \mathcal{B}} [W_2[(\mathbf{z}_{t+1})]_{\mathbb{K}}] \|_D \\
&\leq \gamma \cdot \max_{\mathbb{K} \in \mathcal{K}} \sup_{\mathbf{z} \in \mathcal{B}} \|\mathbb{E}_{\mathbf{z}_{t+1} \sim \mathcal{B}} [[W_1(\mathbf{z}_{t+1})]_{\mathbb{K}} - [W_2(\mathbf{z}_{t+1})]_{\mathbb{K}}] \|_D \\
&\leq \gamma \cdot \max_{\mathbb{K} \in \mathcal{K}} \sup_{\mathbf{z} \in \mathcal{B}} \|[W_1(\mathbf{z}_{t+1}) - W_2(\mathbf{z}_{t+1})]_{\mathbb{K}} \|_D \\
&= \gamma \cdot \|W_1 - W_2\|_{\mathcal{W}},
\end{aligned}
\tag{20}
$$

*where $\Gamma = diag(\gamma, \gamma, \cdots, \gamma)_{\mathcal{K} \times \mathcal{K}}$ and $\gamma \in [0, 1)$, proving that $\mathcal{F}$ is a contraction mapping.*

## A.3 PROOF OF THE POLICY PERFORMANCE DISTINCTION VIA WAVELET DOMAIN FEATURES

**Lemma 2** *Following the previous definition of state distribution Achiam et al. (2017), we define the discounted task representation distribution as follows:*

$$\mathcal{D}^\pi(z) = (1 - \gamma) \sum_{t=0}^{\infty} \gamma^t \sum_{h=0}^{\infty} \mathcal{P}_{\omega_h}(z_t = z|\pi, y)\rho(\omega_h|\omega_{0:h-1}). \tag{21}$$

*The neural network $y$ maps the trajectory $\tau$ to the task representation, denoted as $z$. Then the expected discounted total reward under policy $\pi$ can be expressed as*

$$
\begin{aligned}
J_\pi &= \sum_{t=0}^{\infty} \gamma^t E_{\omega_h \sim \rho, \tau \sim \pi} \left[ \mathcal{R}_{\omega_h}(\tau) \right] \\
&= \sum_{t=0}^{\infty} \gamma^t \sum_{h=0}^{\infty} \int_\tau \mathcal{R}^\pi(\tau)\mathcal{P}_{\omega_h}(\tau|\pi)\rho(\omega_h|\omega_{0:h-1}) \, \mathrm{d}\tau \\
&= \int_\tau \mathcal{R}^\pi(\tau) \sum_{t=0}^{\infty} \gamma^t \sum_{h=0}^{\infty} \mathcal{P}_{\omega_h}(\tau|\pi)\rho(\omega_h|\omega_{0:h-1}) \, \mathrm{d}\tau \\
&= \int_{z \sim y(\tau)} \mathcal{R}^\pi(z) \sum_{t=0}^{\infty} \gamma^t \sum_{h=0}^{\infty} \mathcal{P}_{\omega_h}(z|\pi)\rho(\omega_h|\omega_{0:h-1}) \, \mathrm{d}z \\
&= \frac{1}{1 - \gamma} \int_{\mathcal{Z}} \mathcal{R}^\pi(z)\mathcal{D}^\pi(z) \, \mathrm{d}z \\
&\stackrel{z \sim y(s,a,s')}{=\!=\!=\!=\!=\!=} \frac{1}{1 - \gamma} E_{\substack{z \sim \mathcal{D}^\pi, a \sim \pi(\cdot|s,z) \\ s' \sim \mathcal{P}(\cdot|s,a,z)}} \left[ \mathcal{R}(z) \right].
\end{aligned}
\tag{22}
$$

**Theorem 2** *Suppose that the reward function $\mathcal{R}(z)$ can be expanded into a Bth-degree Taylor series for $z \in \mathbb{R}^D$, then for any two policies $\pi_1$ and $\pi_2$, their performance difference is bounded as:*

$$|J_{\pi_1} - J_{\pi_2}| \leq \frac{\sqrt{D}}{1-\gamma} \cdot \sum_{b=1}^{B} \frac{\left\| \mathcal{R}^{(b)}(0) \right\|_D}{b!} \cdot \max_{1 \leq q \leq D} \sup_{d_q \in \mathbb{R}, \beta_q > 0} \left| \mathcal{W}_{\pi_1}^{(b)}(d_q, \beta_q) - \mathcal{W}_{\pi_2}^{(b)}(d_q, \beta_q) \right|, \quad (23)$$

*where $\mathcal{W}_\pi^{(b)}(d, \beta)$ denotes the wavelet transform of the bth power of the task representation sequence $\mathbf{z}^{(b)} = [z_0, z_1, \ldots, z_{T_H}]^{(b)}$ for any integer $b \in [1, B]$, and $J_\pi$ represents the average policy performance over the entire task space within a single episode.*

**Proof 3** *First, the reward function can be written as $\mathcal{R}(z) = \sum_{b=0}^{B} \frac{\mathcal{R}^{(b)}(0)^{\mathsf{T}}}{b!} z^b$ based on the Taylor series expansion Ye et al. (2024). According to lemma 2, for any integer $b \in [1, B]$, we have*

$$|J_{\pi_1} - J_{\pi_2}| \leq \frac{1}{1-\gamma} \int_{\mathcal{Z}} \left[ \mathcal{R}^{\pi_1}(z)\mathcal{D}^{\pi_1}(z) - \mathcal{R}^{\pi_2}(z)\mathcal{D}^{\pi_2}(z) \right] \, \mathrm{d}z$$

$$\leq \sum_{b=0}^{B} \frac{\left\| \mathcal{R}^{(b)}(0) \right\|_D}{b!} \cdot \left\| \int_{\mathcal{Z}} \left[ z^b \mathcal{D}^{\pi_1}(z) - z^b \mathcal{D}^{\pi_2}(z) \right] \mathrm{d}z \right\|_D$$

$$= \sum_{b=0}^{B} \frac{\left\| \mathcal{R}^{(b)}(0) \right\|_D}{b!} \left\| \mathop{E}_{z \sim \mathcal{D}^{\pi_1}} \left[ z^b \right] - \mathop{E}_{z \sim \mathcal{D}^{\pi_2}} \left[ z^b \right] \right\|_D. \quad (24)$$

*Since the inverse wavelet transform Burrus et al. (1998) of $\mathcal{W}_\pi^{(b)}(d, \beta)$ is the reconstructed task representation sequence $\mathbf{z}^{(b)}$, we have*

$$\mathop{E}_{z_q \sim \mathcal{D}^\pi} \left[ z_q^b \right] = \int_0^\infty \int_{-\infty}^\infty \mathcal{W}_\pi^{(b)}(d_q, \beta_q) \tilde{\psi} \left( \frac{t - d_q}{\beta_q} \right) \mathrm{d}d_q \mathrm{d}\beta_q, \quad \forall q = 1, 2, \ldots, D, \quad (25)$$

*where $\tilde{\psi}$ is the dual function form of the mother wavelet function $\psi$ and has the normalized orthogonal property. The dimensions of translation position $d$ and scaling factor $\beta$ are identical to those of $z$.*

*Then we have the following derivation:*

$$\left| \mathop{E}_{z_q \sim \mathcal{D}^{\pi_1}} \left[ z_q^b \right] - \mathop{E}_{z_q \sim \mathcal{D}^{\pi_2}} \left[ z_q^b \right] \right| \leq \int_0^\infty \int_{-\infty}^\infty \left| \mathcal{W}_{\pi_1}^{(b)}(d_q, \beta_q) - \mathcal{W}_{\pi_2}^{(b)}(d_q, \beta_q) \right| \cdot \left| \tilde{\psi} \left( \frac{t - d_q}{\beta_q} \right) \right| \mathrm{d}d_q \mathrm{d}\beta_q$$

$$\leq \sup_{d_q \in \mathbb{R}, \beta_q > 0} \left| \mathcal{W}_{\pi_1}^{(b)}(d_q, \beta_q) - \mathcal{W}_{\pi_2}^{(b)}(d_q, \beta_q) \right| \int_0^\infty \int_{-\infty}^\infty \left| \tilde{\psi} \left( \frac{t - d_q}{\beta_q} \right) \right| \mathrm{d}d_q \mathrm{d}\beta_q$$

$$\leq \sup_{d_q \in \mathbb{R}, \beta_q > 0} \left| \mathcal{W}_{\pi_1}^{(b)}(d_q, \beta_q) - \mathcal{W}_{\pi_2}^{(b)}(d_q, \beta_q) \right|. \quad (26)$$

*Then the policy performance difference bound is derived as follows:*

$$|J_{\pi_1} - J_{\pi_2}| \leq \frac{1}{1-\gamma} \cdot \sqrt{D} \cdot \sum_{b=1}^{B} \frac{\left\| \mathcal{R}^{(b)}(0) \right\|_D}{b!} \cdot \max_{1 \leq q \leq D} \sup_{d_q \in \mathbb{R}, \beta_q > 0} \left| \mathcal{W}_{\pi_1}^{(b)}(d_q, \beta_q) - \mathcal{W}_{\pi_2}^{(b)}(d_q, \beta_q) \right|,$$

*which concludes the proof of Theorem 2.*

### A.4 PROOF OF THE POLICY IMPROVEMENT WITH WAVELET TASK REPRESENTATIONS

Suppose that the function class for the contextual policy $\pi$ of WISDOM is defined as $\Pi_{\text{WISDOM}} \doteq \{\pi, \text{s.t. } \exists \theta \text{ with } \pi(\cdot|s, \hat{z}) = f_\theta(s, Y_\phi(e_\eta(\mathcal{C}))), \forall s\}$, where $f_\theta$ is a neural network characterizing the policy conditioned on top of the context encoder $e_\eta$ and the wavelet representation network $Y_\phi$.

**Assumption 1** *(1.1) Access to the true history policy: $\pi_{\hat{h}} = \pi_h$. (1.2) The context encoder $e_\eta$ and the wavelet representation network $Y_\phi$ allows to represent the estimate of the history policy : $\pi_{\hat{h}}(a|s, Y_\phi(e_\eta(\mathcal{C}))) = \pi_{\hat{h}}(a|s, \hat{z}) \in \Pi_{WISDOM}$. (1.3) Access to the true performance of policies: $\hat{J}(\pi) = J(\pi)$ for all policies $\pi$. (1.4) The non-stationary RL algorithm performs perfect optimization on top of $e_\eta$ and $Y_\phi$: $J_{WISDOM} = \max_{\pi \in \Pi_{WISDOM}} \hat{J}(\pi)$.*

**Theorem 3** *WISDOM returns a policy $\pi(\cdot|s, \hat{z})$ conditioned on the wavelet task representation that improves upon its intermediate policy produced during policy iteration $\pi_h$, satisfying $J_{WISDOM} \geq J_{\pi_h}$ under Assumption 1.1-1.4, where $J_{WISDOM}$ denotes the final policy performance of WISDOM.*

**Proof 4**

$$
\begin{aligned}
J_{WISDOM} &= J\left(\arg\max_{\pi \in \Pi_{WISDOM}} \hat{J}(\pi)\right) &\text{(Assumption 1.4)}\\
&= \max_{\pi \in \Pi_{WISDOM}} J(\pi) &\text{(Assumption 1.3)}\\
&\geq J\left(\pi_{\hat{h}}\right) &\text{(Assumption 1.2)}\\
&\geq J\left(\pi_h\right) &\text{(Assumption 1.1)}
\end{aligned}
$$

*which concludes the proof of Theorem 3.*

## B  EXPERIMENTAL DETAILS

### B.1  BASELINES

We compared WISDOM with seven prevailing and competitive baselines. Initially, SAC, a standard RL method, serves as the backbone to validate the importance of task inference. Conversely, PEARL and $RL^2$, which are classical meta-RL methods, perform task inference using an encoder, emphasizing the significance of modeling the task evolution process in non-stationary environments. Moreover, the last four baselines (CEMRL, SeCBAD, TRIO, and COREP) are specifically designed to address RL tasks in non-stationary settings, further substantiating the advantages of modeling and predicting the task evolution process through wavelet transform in WISDOM. Their introductions are as follows:

(1) **SAC** Haarnoja et al. (2018) is a standard model-free, off-policy reinforcement learning method that aims to maximize both the expected return and the entropy of the policy;

(2) **PEARL** Rakelly et al. (2019) is a classical context-based meta-RL method that enhances sample efficiency and policy performance by disentangling task inference and control, employing online probabilistic filtering of latent task variables for effective adaptation to new tasks from limited experience;

(3) **$RL^2$** Duan et al. (2016) employs RNNs to encode states for meta-learning, storing learned prior knowledge in the hidden states. This prior knowledge is then applied across multiple tasks, and RL algorithms are used to train the weights of the RNN, addressing the challenge of rapid learning;

(4) **CEMRL** Bing et al. (2023b) leverages a Gaussian Mixture Model to represent non-stationary task distributions with cluster structure and learns a more compact latent space through reconstructing tasks;

(5) **SeCBAD** Chen et al. (2022) perceives and adapts to new evolving tasks through reward functions, leading to a policy that can adapt to rapid variations;

(6) **TRIO** Poiani et al. (2021) meta-trains a variational module for inferring distributions over latent context and leverages a Gaussian Process (GP) to model the task evolution process;

(7) **COREP** Zhang et al. (2024a) utilizes a dual graph structure to retrospect causal origin of non-stationarity, with a core graph emphasizing stable representation and a general graph compensating for overlooked edge information.

For fair comparison, WISDOM and all baseline models, except TRIO and $RL^2$ which emphasize the use of a recurrent encoder, employed the same MLP network architecture for their context encoders.

### B.2  EVALUATION METRICS

All models were trained with equal time steps, and subsequent policy evaluations were performed with the same number of time steps. For evaluating the policy's performance on Meta-World, the average success rate was computed across six random seeds. On MuJoCo and Type-1 Diabetes control, the average return was computed across six random seeds.

### B.3 NON-STATIONARY EXPERIMENTAL SETTINGS

We evaluated our proposed model WISDOM on various non-stationary tasks with uncertain evolution periods, using three benchmarks: **Meta-World**, **Type-1 Diabetes** and **MuJoCo**. Then we describe the details of these three environments.

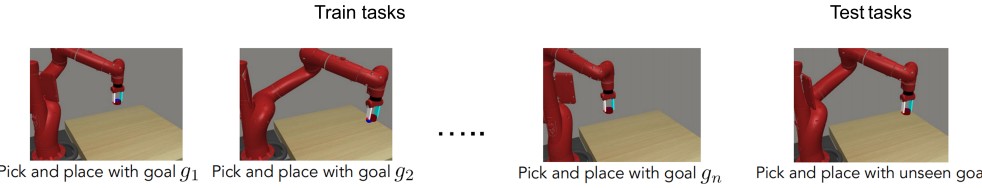

Figure 9: Visualization of ML1 evaluation protocol.

**Meta-World**  Meta-World is an open-source simulated benchmark for meta-RL and multi-task learning, consisting of 50 distinct robotic manipulation tasks. As shown in Fig. 9, we experiment with the Meta-Learning 1 (ML1) evaluation protocol, which evaluates meta-RL algorithms' few-shot adaptation to goal variation within one task. ML1 uses single Meta-World tasks, with the meta-training tasks corresponding to 50 random initial object and goal positions, and meta-testing conducted on 10 held-out positions. The non-stationarity in Meta-World is evidenced by the continuous fluctuation of the target position for the robotic arm's movement, which is correlated with the reward function. We selected 8 different types of tasks, and the detailed descriptions are listed in Table 2.

Table 2: Meta-World task descriptions.

| Task | Description |
| --- | --- |
| Button-Press | Randomize button positions and press a button. |
| Door-Close | Randomize door positions and close a door with a revolving joint. |
| Door-Lock | Randomize door positions and lock the door by rotating the lock clockwise. |
| Door-Unlock | Randomize door positions and unlock the door by rotating the lock counter-clockwise. |
| Faucet-Close | Randomize faucet positions and rotate the faucet clockwise. |
| Handle-Press | Randomize the handle positions and press a handle down. |
| Plate-Slide | Randomize the plate and cabinet positions and slide a plate into a cabinet. |
| Plate-Slide-Back | Randomize positions for both the plate and cabinet retrieve a plate from the cabinet. |

**Type-1 Diabetes**  In addition to the MuJoCo and Meta-World physical control tasks, we also evaluate our proposed approach on a more challenging real-world task: Realistic Type-1 Diabetes, which aims to control an insulin pump to regulate the blood glucose level of a patient. Each step of an episode corresponds to a minute in an in-silico patient's body, and each episode consists of 1440 time steps, corresponding to one day. When a patient consumes food, the blood glucose level will increase. Extremely high or low blood glucose levels can lead to life-threatening conditions such as hyperglycemia and hypoglycemia, respectively. Therefore, insulin dosage is required to mitigate the risks associated with these conditions.

In the glucose control task, the states are hidden patient parameters, the observations are continuous blood glucose readings, the actions consist of discrete insulin dosages within the range [0, 5], and the transition dynamics are specific to the patient but unknown to the agent. Following the previous work Basu et al. (2023), we design a biologically inspired custom zone reward that incentivizes the time spent in the target zone and penalizes hyperglycemia and hypoglycemia:

$$r_t(s_{t-1}, s_t) = \begin{cases} -20 & s_t < 70 \text{ or } s_t > 200 \ (\text{episode termination}) \\ -1 & s_t < 100 \text{ or } s_t - s_{t-1} < 0.5 \ (\text{hypoglycemia}) \\ -1 & s_t > 150 \text{ or } s_t - s_{t-1} > 0.5 \ (\text{hyperglycemia}) \\ 50 & 100 \le s_t \le 150 \ (\text{target blood glucose}) \end{cases} \tag{27}$$

where $r_t$ encourages the agent to maintain the target vital statics within a healthy range.

We induced non-stationarity by oscillating the body parameters, such as the consumption amount of total glucose. Specifically, we simulate non-stationary fluctuations in blood glucose levels by varying the daily intake of lunch (1 meal) or both lunch and dinner (2 meals) in both adolescent and

adult populations. Specifically, the meal plan (meal time and size) is set as follows: 7:00 AM: 45 (breakfast), 12:00 AM: 70 (launch), 16:00 PM: 15 (snack1), 18:00 PM: 80 (dinner), 23:00 PM: 10 (snack). The meal plan for adults is exactly the same as that for adolescents. The range for the size of the adults' meals for lunch or lunch and dinner is 60-80, while the range for adolescents for lunch or lunch and dinner is 50-80. The goal of this system is to responsibly update the doctor's initial prescription, ensuring that treatment continually improves.

**MuJoCo** We also adopted 4 classic MuJoCo control tasks for comparison. In the setting of non-stationary RL, we primarily considered two different types of MuJoCo evolution tasks: changes in reward functions and dynamics. The environment details and the number of tasks for meta-training and meta-testing in different tasks are shown in Table 3.

- **Changes in reward functions.** In the Hopper-Vel and Walker-Vel tasks, the target velocity of the agent is sampled from the uniform distribution $U[0.5, 3.0]$ every $60 \pm 20$ time steps to simulate non-stationarity.

- **Changes in dynamics.** In Cheetah-Damping, the damping parameters are sampled from the set $[0.85, 0.9, 0.95, 1.0]$. In Walker-Rand-Params, the physical parameters of the agent, such as body mass, damping, and friction, are randomized. The variation in these physical parameters at $60 \pm 20$ time steps results in non-stationary environmental dynamics.

Table 3: Classical MuJoCo details.

| Task | Observation dim | Action dim | Training tasks | Test tasks |
|------|-----------------|------------|----------------|------------|
| Cheetah-Damping | 17 | 6 | 60 | 10 |
| Hopper-Vel | 11 | 3 | 100 | 30 |
| Walker-Rand-Params | 17 | 6 | 100 | 30 |
| Walker-Vel | 17 | 6 | 100 | 30 |

**Hyperparameters** The hyperparameters utilized in the experiments are outlined in Table 4.

Table 4: Hyperparameters.

| Hyperparameter | Value |
|----------------|-------|
| Encoder training steps | 200 |
| RL layer size | 300 |
| Target entropy factor | 1.0 |
| Learning rate | 3e-4 |
| Dims. of task representation $z$ | 5 |
| Coef. of soft update $\sigma$ | 5e-3 |
| Replay buffer size | 10,000,000 |
| Batch size of the policy | MuJoCo & Type-1 Diabetes: 256, Meta-World: 512 |
| Batch size of the encoder | MuJoCo & Type-1 Diabetes: 256, Meta-World: 512 |
| Maximum trajectory length | MuJoCo & Type-1 Diabetes: 800, Meta-World: 200 |
| Evaluation trajectories | MuJoCo & Type-1 Diabetes: 2, Meta-World: 1 |
| Policy training steps | MuJoCo: 2,000, Meta-World: 4,000, Type-1 Diabetes: 200 |
| Training tasks per episode | MuJoCo (Cheetah: 20), Type-1 Diabetes: 25, Meta-World: 50 |
| Initial trainsitions | MuJoCo & Meta-World: 200, Type-1 Diabetes: 400 |
| Trainsitions per episode | MuJoCo: 800, Meta-World: 600, Type-1 Diabetes: 200 |
| Decomposition level $M$ | MuJoCo (Walker-Rand-Params: 3), Type-1 Diabetes: 2 Meta-World (Button-Press & Handle-Press: 5): 2 |
| Coef. of wavelet TD loss $\alpha_Y$ | MuJoCo (Hopper-Vel: 0.5): 0.9 Type-1 Diabetes (Adolescent-1meal: 0.5): 0.1 Meta-World (Faucet-Close & Door-Close: 0.9, Door-Lock: 0.8): 0.1 |

## C PSEUDOCODE

---
**Algorithm 1** WISDOM algorithm
---

**Input**: training tasks $\tilde{\mathcal{T}}^{train}$ from $\mathcal{P}(\tilde{\mathcal{T}})$, replay buffer $\mathcal{B}$, context encoder $e_\eta(\mathbf{z}|\mathcal{C})$, wavelet representation network $Y_\phi(\hat{\mathbf{z}}|\mathbf{z})$, $W$ network $W_\varphi(\mathbf{u}|\mathcal{C}, e)$, entropy term $\mathcal{H}$, contextual policy $\pi_\theta(a|s, \hat{z})$, critic $Q_\upsilon(s, a, \hat{z})$.

---

**Collecting Training Data**

**for** training task $\tilde{\mathcal{T}}^{train}$ **do**
    Generate a task representation sequence $\mathbf{z} \sim e_\eta(\mathbf{z}|\mathcal{C})$
    Roll-out policy $\pi_\theta(a|s, z)$ and add transitions $(s_j, a_j, s'_j, r_j)_{j:1\ldots\mathbb{J}}$ to $\mathcal{B}$
**end for**

---

**Training Context Encoder and Wavelet Representation Network**

**for** each context encoder training step **do**
    Sample $c_i = (s_i, a_i, s'_i, r_i)_{i:1\ldots\mathbb{I}} \sim \mathcal{B}$ for context encoder
    Generate a task representation $z \sim e_\eta(z|c_i)$
    Obtain a wavelet task representation sequence: $\hat{\mathbf{z}} = Y_\phi(\mathbf{z})$
    Train context encoder $e_\eta$: $\eta \leftarrow \eta - \lambda_\eta \hat{\nabla}_\eta \mathcal{J}_\eta$ (Eq. 3)
    Train wavelet representation network $Y_\phi$: $\phi \leftarrow \phi - \lambda_\phi \hat{\nabla}_\phi \mathcal{J}_\phi$ (Eq. 5)
    Train $W$ network $W_\varphi$: $\varphi \leftarrow \varphi - \lambda_\varphi \hat{\nabla}_\varphi \mathcal{J}_\phi$ (Eq. 5)
    Train target $W$ network $W_\mu$: $\mu \leftarrow \sigma\varphi + (1 - \sigma)\mu$
**end for**

---

**Training Policy**

**for** each policy training step **do**
    Train contextual policy $\pi_\theta$: $\theta \leftarrow \theta - \lambda_\pi \hat{\nabla}_\theta \mathcal{J}_\theta$ (Eq. 7)
    Obtain a wavelet task representation sequence: $\hat{\mathbf{z}} = Y_\phi(\mathbf{z})$
    Train contextual critic $Q_\upsilon$: $\upsilon_l \leftarrow \upsilon_l - \lambda_Q \hat{\nabla}_{\upsilon_l} \mathcal{J}_{\upsilon_l}$, for $l \in \{1, 2\}$ (Eq. 6)
    Train temperature coefficient $\alpha$: $\alpha \leftarrow \alpha - \lambda_\alpha \hat{\nabla}_\alpha \mathbb{E}\left[-\alpha \log \pi_\theta(a|s, \hat{z}) - \alpha\mathcal{H}\right]$
    Train target contextual critic $Q_\zeta$: $\zeta_l \leftarrow \sigma\upsilon_l + (1 - \sigma)\zeta_l$, for $l \in \{1, 2\}$
**end for**

---

**Testing on Unseen Tasks**

Initialize transitions $c^{\tilde{\mathcal{T}}} = \{\}$
Sample test tasks $\tilde{\mathcal{T}}^{test}$ from $\mathcal{P}(\tilde{\mathcal{T}})$
**for** $g = 1, \ldots, \mathbb{G}$ **do**
    Generate a task representation $z \sim e_\eta(z|c_g)$
    Roll out policy $\pi_\theta(a|s, z)$ to generate transitions $\mathbb{D}_g^{\tilde{\mathcal{T}}} = (s_j, a_j, s'_j, r_j)_{j:1\ldots\mathbb{J}}$
    Store the transitions: $c^{\tilde{\mathcal{T}}} = c^{\tilde{\mathcal{T}}} \cup \mathbb{D}_g^{\tilde{\mathcal{T}}}$
**end for**

---

## D IMPLEMENTATION DETAILS

**Model Architecture** All components of WISDOM are implemented as MLPs. We report the total number of learnable parameters for our model initialized for the Walker-Vel task ($\mathcal{S} \in \mathbb{S}^{17}$, $\mathcal{A} \in \mathbb{A}^6$). We summarize the architecture and model size of WISDOM using PyTorch-like notation in Alg. 2.

**Training Details** We evaluate our proposed method and other baselines on the NVIDIA GeForce RTX 3090 GPU. And PyTorch-like pseudo-code for training our model is illustrated in Alg. 3.

**Computational Cost** WISDOM adds only a modest time cost. As depicted in Table 5, WISDOM stands out with the lowest memory usage (Mb) among all baselines and a shorter runtime per episode (s) compared to competitive NSRL methods such as SeCBAD and COREP. WISDOM is also more efficient than the classical meta-RL method PEARL, despite PEARL not modeling task changes.

---

**Algorithm 2** WISDOM Network Structure Pytorch-like Pseudocode

---

```
Encoder parameters: 90,810
Q1, Q2, target Q1, target Q2 parameters: 189,601
Policy parameters: 191,112
Alpha network parameters: 351

Encoder(
  (fc0): Linear(in_features=2 * o_dim + a_dim + r_dim, out_features=200, bias=True)
  (fc1): Linear(in_features=200, out_features=200, bias=True)
  (fc2): Linear(in_features=200, out_features=200, bias=True)
  (last_fc): Linear(in_features=200, out_features=10, bias=True)
)

Q1, Q2, target Q1, target Q2(
  (fc0): Linear(in_features=(o_dim + latent_dim) + a_dim, out_features=300, bias=True)
  (fc1): Linear(in_features=300, out_features=300, bias=True)
  (fc2): Linear(in_features=300, out_features=300, bias=True)
  (last_fc): Linear(in_features=300, out_features=1, bias=True)
)

Policy(
  (fc0): Linear(in_features=22, out_features=300, bias=True)
  (fc1): Linear(in_features=300, out_features=300, bias=True)
  (fc2): Linear(in_features=300, out_features=300, bias=True)
  (last_fc): Linear(in_features=300, out_features=6, bias=True)
  (last_fc_log_std): Linear(in_features=300, out_features=6, bias=True)
)

Alpha network(
  (fc0): Linear(in_features=5, out_features=50, bias=True)
  (last_fc): Linear(in_features=50, out_features=1, bias=True)
)
```

---

**Algorithm 3** WISDOM Training Pytorch-like Pseudocode

---

```
def train(self):

    # Collecting initial samples ...
    if self.initial_transitions > 0:
        self.env_steps += self.collect_data(self.train_tasks, self.initial_transitions)

    for epoch in gt.timed_for(range(self.num_epochs), save_itrs=True):

        # 1. Collect data with rollout coordinator
        collection_tasks = np.random.permutation(self.train_tasks)[:self.num_train_tasks]
        self.env_steps += self.collect_data(collection_tasks, self.num_transitions)

        # 2. Replay buffer stats
        self.replay_buffer.stats_dict = self.replay_buffer.get_stats()

        # 3. Train context encoder and wavelet representation network
        self.reconstruction_trainer.train(self.num_reconstruction_steps)

        # 4. Train policy with data from the replay buffer
        temp, sac_stats = self.policy_trainer.train(self.num_policy_steps)

        # 5. Evaluation
        eval_output = self.evaluate(LOG, 'train', collection_tasks, self.eval_traj)
        eval_output = self.evaluate(LOG, 'test', self.test_tasks, self.eval_traj)
        average_test_reward, std_test_reward = eval_output
```

---

Table 5: Computational cost comparison among different methods.

| Task | WISDOM | PEARL | SeCBAD | RL$^2$ | TRIO | COREP | CEMRL |
|---|---|---|---|---|---|---|---|
| Button-Press | **432M** | 455M | 457M | 459M | 499M | 695M | 741M |
| | 251.32s | 255.96s | 409.86s | 185.84s | **170.66s** | 1207.51s | 171.44s |
| Walker-Vel | **406M** | 421M | 425M | 449M | 443M | 501M | 787M |
| | 175.34s | 183.21s | 285.37s | **80.12s** | 108.05s | 348.94s | 327.98s |

# E    ADDITIONAL EXPERIMENT RESULTS

## E.1    ADDITIONAL ABLATION RESULTS

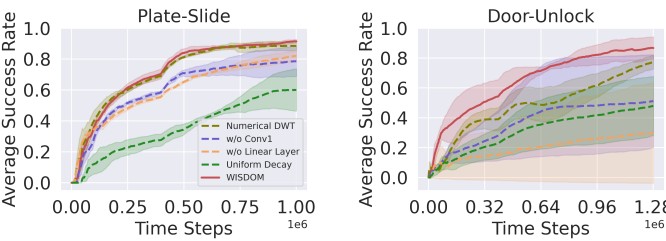

Figure 10: Ablation on the design of the wavelet representation network.

## E.2    EVALUATION RESULTS ACROSS DIVERSE NON-STATIONARY DEGREES

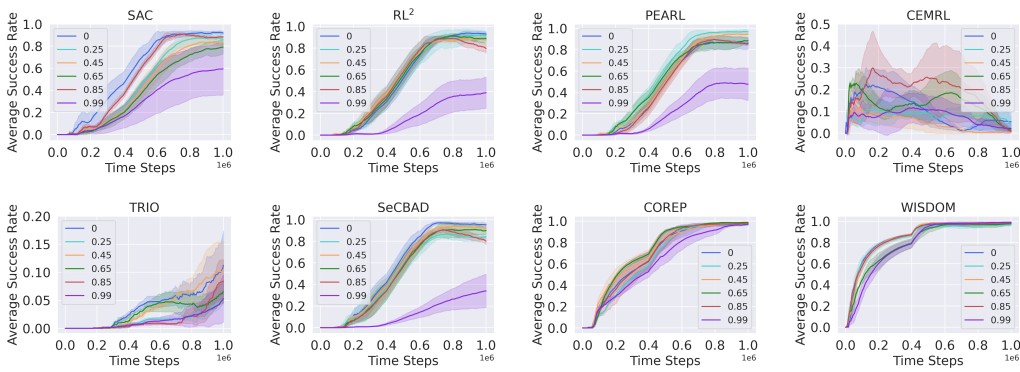

Figure 11: Testing average performance on button-press with varying non-stationary degrees.

## E.3    PERFORMANCE DISPARITIES OF META-RL IN STATIONARY AND NON-STATIONARY TASKS

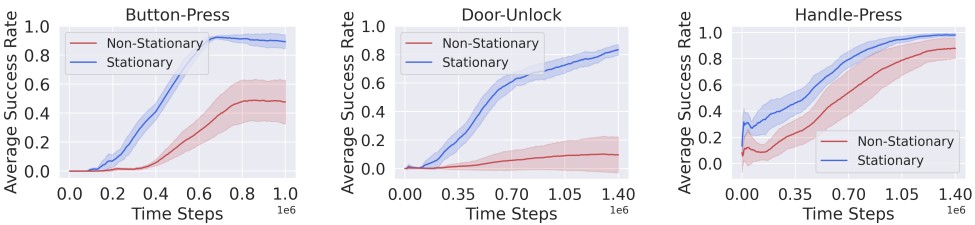

Figure 12: Testing performance of the classical meta-RL algorithm PEARL Rakelly et al. (2019) in stationary and non-stationary environments on Meta-World Yu et al. (2020).

## E.4    SENSITIVITY ANALYSIS OF WAVELET DECOMPOSITION LEVELS AND WAVELET TD LOSS COEFFICIENTS

An appropriate setting of wavelet decomposition levels $M$ allows for more precise separation of frequency components corresponding to distinct evolutionary periods, thereby facilitating the extraction of more critical features. In Fig. 13 (left), increasing $M$ moderately can filter out task-irrelevant information and improve adaptability. However, excessive decomposition may lose vital low-frequency components, thereby hindering the learning of non-stationary trends. In Fig. 13 (right), $\alpha_Y$ typically influences the convergence speed. When wavelet TD loss or AR loss predominates, the convergence accelerates; even so, the final performance remains insensitive to $\alpha_Y$.

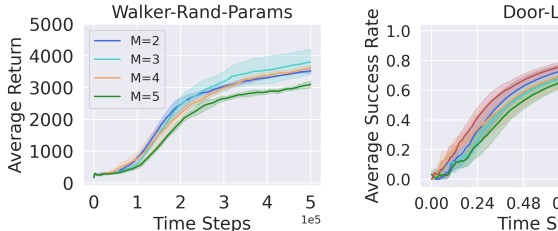

Figure 13: Sensitivity to wavelet decomposition levels $M$ (left) and TD loss coefficients $\alpha_Y$ (right).

## E.5   ADDITIONAL EXPERIMENTAL RESULTS ON MUJOCO BENCHMARK

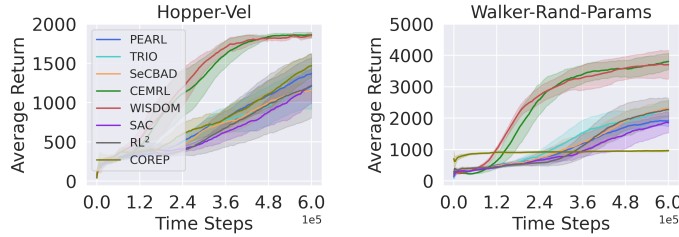

Figure 14: Testing average return on MuJoCo over 6 random seeds.

## E.6   ADDITIONAL EXPERIMENTAL RESULTS ON GLUCOSE CONTROL BENCHMARK

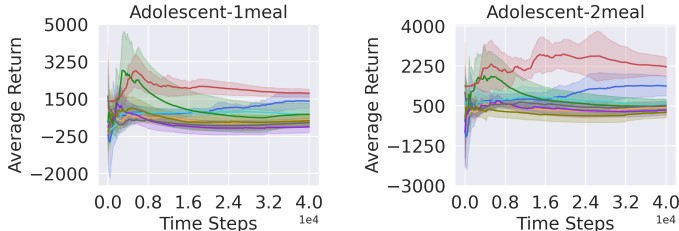

Figure 15: Testing average return on 2 glucose control environments over 6 random seeds.

## E.7   EVALUATION ON MORE DIFFICULT TASK

We conducted additional experiments on more complex Reach, Door-Open, and Hammer to evaluate the performance of WISDOM. As illustrated in Fig. 16, WISDOM consistently outperforms popular non-stationary baselines on these more challenging tasks, indicating that wavelet-based task representations can effectively capture the underlying trends of complex non-stationary changes and remain robust across a broader range of task difficulties.

## E.8   SENSITIVITY ANALYSIS OF WAVELET BASIS FUNCTIONS

We evaluated the sensitivity of WISDOM to the choice of wavelet basis by conducting additional experiments on Button-Press using three representative wavelet bases (Bior1.1, Sym2, and Haar). As illustrated in Fig. 17, different bases lead to slight differences in convergence speed, but they have minimal impact on the final performance. In our wavelet representation network, the wavelet basis functions serve only as an initialization to approximate traditional wavelet behavior. Because the filters are learnable, the model progressively reduces reliance on the initial basis and is able to extract more expressive and task-adaptive features from non-stationary task sequences without relying on hand-crafted designs or a fixed wavelet basis. This explains why different bases influence convergence only in the early stages while having a minimal impact on the final performance.

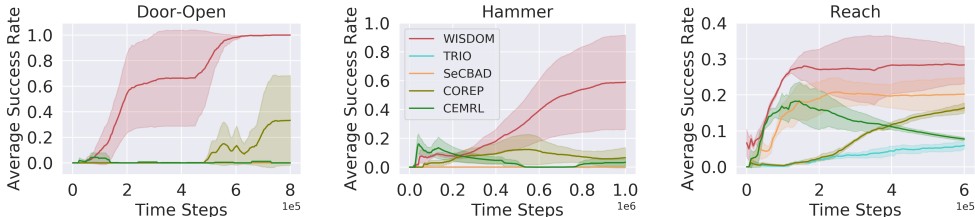

Figure 16: Testing success rate on more difficult tasks over 3 random seeds.

Among the three wavelet bases, Sym2 converges the fastest, followed by Haar, whereas Bior1.3 converges the slowest. This can be attributed to the strict orthogonality of the Haar wavelet, which reduces feature redundancy and enhances learning efficiency, leading to faster convergence compared to the biorthogonal Bior1.3. The longer filters of Bior1.3 incur greater computational cost and slower convergence. In contrast, the Sym2 wavelet, with its smooth and near-Gaussian shape, more effectively captures stable trends and suppresses high-frequency noise, thereby accelerating convergence and yielding superior final performance. Overall, the comparable results across different bases demonstrate that WISDOM is largely insensitive to the choice of wavelet basis functions.

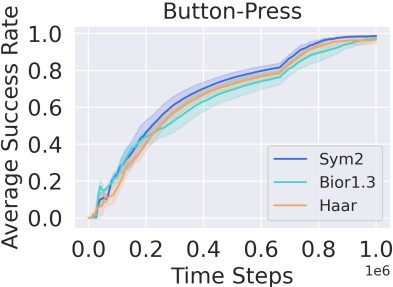

Figure 17: Sensitivity analysis of different wavelet basis functions.

### E.9 EVALUATION ON TASK WITH SPARSE REWARD

Following the sparse-reward setup in LatCo Rybkin et al. (2021), the agent receives a reward only when it successfully completes the task. As shown in Fig. 18, WISDOM consistently outperforms the baselines, and the wavelet TD update further enhances both adaptability and stability. This demonstrates that our wavelet representation can handle non-stationary tasks with sparse reward signals, enabling rapid adaptation. Importantly, the explicit wavelet TD update helps the learned representations become denser and more informative, while also highlighting low-reward yet critical features. Thus, WISDOM also exhibits exceptional adaptation efficiency on tasks with sparse rewards.

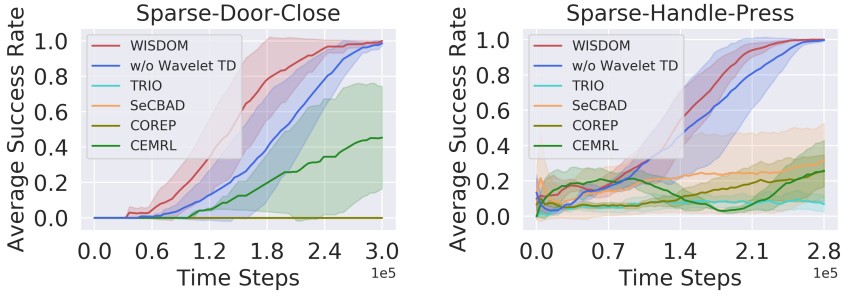

Figure 18: Testing success rate on sparse tasks over 3 random seeds.

### E.10 ADDITIONAL EVALUATION ON TASK WITH TRANSITION DISTRIBUTION DRIFTS

We conducted an additional experiment on Walker-Mass to further evaluate the wavelet TD update. As depicted in Fig. 19, WISDOM consistently outperforms the baselines, and the wavelet TD update further enhances both adaptability and stability. This demonstrates that our wavelet representation can handle non-stationary changes not only in reward functions but also in state transition dynamics, enabling rapid adaptation.

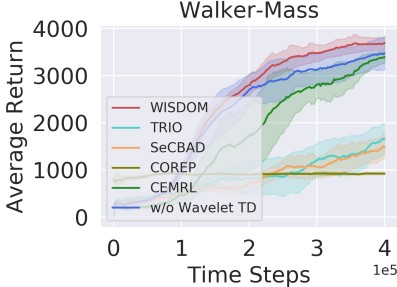

Figure 19: Testing average return on Walker-Mass with transition distribution drifts.

### E.11 ADDITIONAL EVALUATION OF WISDOM'S PLUG-IN CAPABILITY

Our wavelet task representation can be viewed as a plug-in module and can be connected with non-stationary policy methods. We conducted additional experiments to validate that WISDOM can be integrated with online/offline Non-Stationary RL (NSRL) methods as well as model-based RL. The testing curves on unseen tasks are provided in Fig. 20.

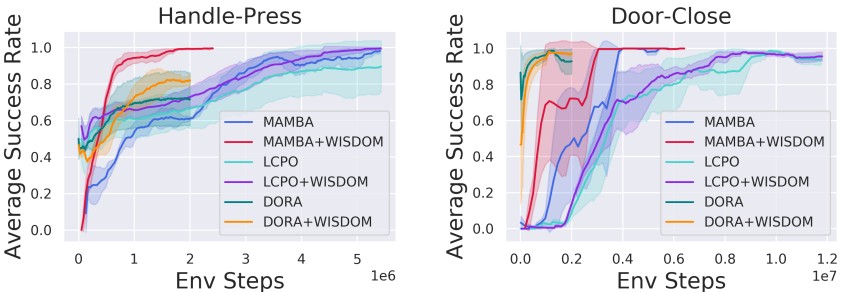

Figure 20: Testing success rate on Non-stationary Handle-Press and Door-Close tasks for validating WISDOM's plug-in capability over 3 random seeds.

LCPO Hamadanian et al. (2023) is an online non-stationary policy learning method that does not incorporate additional representation learning; it mitigates catastrophic forgetting by constraining policy optimization using experience samples that lie outside the current context distribution. Building on LCPO, our wavelet-based task representation further accelerates adaptation and improves the final success rate. This further validates the generality and effectiveness of our wavelet representation and shows that it can facilitate policy improvement, consistent with our analysis in Theorem 3.

DORA Zhang et al. (2024c) is an offline NSRL method that incorporates mutual information to learn task representations capable of differentiating between changes in the environment and the behavior policy. By filtering out spurious high-frequency components and learning stable multi-scale features, our wavelet-based representations reduce misleading correlations in the data and mitigate the distribution shift, effectively preventing policy performance degradation, as observed in Door-Close.

Using the model-based meta-RL method MAMBA Rimon et al. (2024), which is built on a world model, we show that incorporating our wavelet representation significantly improves adaptation efficiency on non-stationary tasks. The potential reason may be that the wavelet representation

explicitly separates the non-stationary changing patterns, making it easier for the world model to capture the true environmental dynamics and thereby improve its prediction accuracy.

### E.12 ADDITIONAL EVALUATION OF WISDOM'S LEARNED REPRESENTATIONS

We conducted experiments using Centered Kernel Alignment (CKA) Kornblith et al. (2019) and Mutual Information Neural Estimation (MINE) Belghazi et al. (2018) metrics to further assess the quality of the learned representations. Using the Inverted Double Pendulum as an example, we measured the mutual information between $\hat{\mathbf{z}}$ and $\mathcal{M}_{\omega_{0:h}}$ with MINE. As shown in Fig. 21(a), as training progresses, the mutual information (polyline) steadily increases alongside the episode return (bar chart), demonstrating a clear positive correlation between their performances. This indicates that the wavelet representation progressively captures the underlying dynamics of the non-stationary tasks and thereby facilitates policy improvement.

In our framework, we utilized two different convolutional layers to learn the approximation and detail coefficients, respectively, followed by the same final linear layer. We computed the representation similarity across layers for the CKA analysis. The heatmaps in both Fig. 21(b) and Fig. 21(c) reveal a clear hierarchical structure, with similarity decreasing as layer distance increases, showing that the model does not collapse into identical mappings. This trend is consistent with observations from the CKA literature Kornblith et al. (2019), where early-layer similarity in well-trained neural networks tends to saturate more easily than in deeper layers.

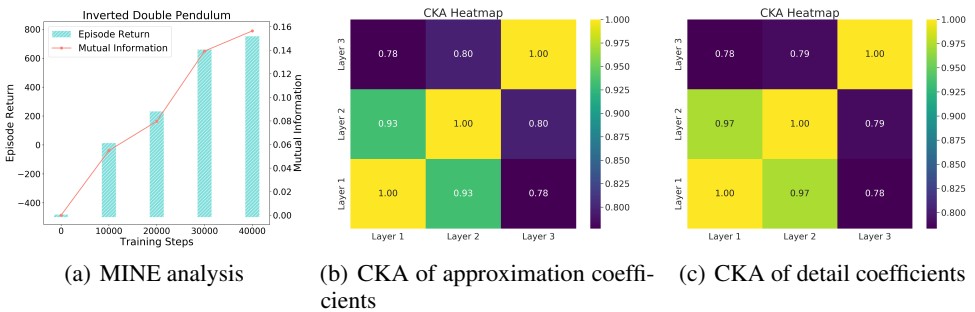

(a) MINE analysis   (b) CKA of approximation coefficients   (c) CKA of detail coefficients

Figure 21: Qualitative analysis of the wavelet task representations.

