# OpenReview forum: "Wavelet Predictive Representations for Non-Stationary Reinforcement Learning"
_ICLR.cc/2026/Conference — ICLR 2026 Poster_

### Official Review · Reviewer_SUaC · 2025-10-21

**Soundness:** 3
**Presentation:** 3
**Contribution:** 2
**Rating:** 4
**Confidence:** 4

**Summary:**

This paper aims to address an important problem: the non-stationary environment in deep reinforcement learning.
To this end, this paper proposes WISDOM to leverage wavelet-domain predictive task representations to enhance non-stationary RL.
Experiments show the effectiveness of the proposed method.

**Strengths:**

1. The paper is overall well-organized and easy to follow.
2. The non-stationary environment in deep reinforcement learning is an important and interesting problem.
3. The authors provide sufficient theoretical support, and the proposed method shows decent performance in numerical experiments.

**Weaknesses:**

1. Compared to existing methods, the proposed method seems to perform decently in the numerical experiments, but the improvement is not that significant, especially for the final performance in Fig.3. Without additional analysis on the quality of the learned representation, it is not clear if the performance benefits indeed come from the proposed wavelet-domain representation.

2. I think the major contribution of this work is the proposed new representation in NSRL. Thus, it would be good to discuss the metrics that can directly evaluate the "quality" or "informativeness" of the learned representations. Such metrics might include Centered Kernel Alignment and Mutual Information Neural Estimation.

3. The time cost of different methods should be given.

**Questions:**

1. Can you connect your work with the non-stationary policy methods?

---

> ### Author Response · Authors · 2025-11-27
> **Responses 1/2**
>
> Thank you for your constructive comments and suggestions. We have revised the manuscript according to your feedback. Below are our responses to your questions, and we would be grateful if you could let us know whether our responses resolve your concerns.
>
> **W1: Clarification on the final performance in Fig.3**
>
> Thanks for the observation. Our WISDOM achieves more significant final performances on more difficult tasks. We conducted additional experiments on more complex Reach, Door-Open, Hammer, and sparse-reward tasks. WISDOM continues to outperform all baselines and achieves significantly higher final performance. The testing curves on unseen tasks are reported in Appendix Fig.16 and Fig.18 of the revised paper, and the corresponding final success rates are shown below:
>
> **Table 1: Converged average test success rate $\pm$ standard error (\%) on more difficult tasks.**
>
> |Task|WISDOM|COREP|SeCBAD|TRIO|CEMRL|
> |-|-|-|-|-|-|
> |Door-Open|**100.00$\pm$0.00**|33.00$\pm$44.64|0.00$\pm$0.00|0.00$\pm$0.00|0.00$\pm$0.00|
> |Hammer|**61.11**$\pm$42.23|6.67$\pm$9.22|0.00$\pm$0.00|0.00$\pm$0.00|2.33$\pm$**3.30**|
> |Reach|**28.00**$\pm$9.09|27.00$\pm$**8.12**|21.67$\pm$9.04|11.67$\pm$9.31|8.33$\pm$9.48|
>
> **Table 3: Final testing performance on tasks with sparse rewards.**
>
> |Task|WISDOM|w/o Wavelet TD|CEMRL|TRIO|SeCBAD|COREP|
> |-|-|-|-|-|-|-|
> |Sparse-Door-Close|**100.00$\pm$0.00**|99.78$\pm$0.31|46.89$\pm$42.63|0.00$\pm$0.00|0.00$\pm$0.00|0.00$\pm$0.00|
> |Sparse-Handle-Press|**100.00$\pm$0.00**|99.65$\pm$0.50|26.14$\pm$24.58|7.02$\pm$9.07|32.63$\pm$30.46|25.61$\pm$16.65|
>
> Since the maximum success rate in Fig.3 is 1.00, some baselines may eventually attain competitive final performance in relatively easy environments, given a sufficiently long adaptation time. However, in the more challenging Door-Unlock and Plate-Slide tasks, where most baselines fail to perform reliably, WISDOM consistently achieves a final success rate of around 0.9, yielding an improvement of approximately 30\%. Notably, across all unseen test tasks in Fig.3, WISDOM exhibits remarkably rapid adaptation, highlighting the strength of wavelet-based representation learning in scenarios that demand efficient online adaptation.
>
> **W1W2: Further evaluation of the quality of the learned representations and their relationship to final performance.**
>
> Thanks for your advice. We conducted new experiments using the reviewer-suggested metrics, including Centered Kernel Alignment (CKA) [1] and Mutual Information Neural Estimation (MINE) [2], to further assess the quality of the learned representations. Using Inverted Double Pendulum as an example, we measured the mutual information between $\mathbf{\hat{z}}$ and $\mathcal{M}_{\omega}$ with MINE. As shown in Appendix Fig.21(a) of the revised paper, as training progresses, the mutual information (polyline) steadily increases alongside the episode return (bar chart), demonstrating a clear positive correlation between their performances. This indicates that the wavelet representation progressively captures the underlying dynamics of the non-stationary tasks and thereby facilitates policy improvement.
>
> In our framework, we utilized two different convolutional layers to learn the approximation and detail coefficients, respectively, followed by the same final linear layer. Inspired by the suggestions provided by the reviewer, we computed the representation similarity across layers for the CKA analysis. The heatmaps in both Fig.21(b) and (c) reveal a clear hierarchical structure, with similarity decreasing as layer distance increases, showing that the model does not collapse into identical mappings. This trend is consistent with observations from the CKA literature [1], where early-layer similarity in well-trained neural networks tends to saturate more easily than in deeper layers.
>
> Besides, the previous ablation studies in Fig.6(a) conducted on the wavelet representation network, along with the updated wavelet TD loss and AR loss, also confirm that the learned wavelet representations indeed contribute to the final performance. Moreover, the previous visualization of learned representations in Fig.8 further demonstrates that the predicted wavelet task representation sequence $\mathbf{\hat{z}}$ intimately aligns with the ground-truth non-stationary task sequence $\mathcal{M}{\omega}$.
>
> [1] Simon et al. Similarity of neural network representations revisited. ICML 2019.
>
> [2] Mohamed et al. Mutual information neural estimation. ICML 2018.

---

> > ### Author Response · Authors · 2025-11-27
> > **Responses 2/2**
> >
> > **W3: Our WISDOM adds only a modest time cost.** Using the Button-Press and Walker-Vel tasks on Meta-World and MuJoCo as examples, WISDOM stands out with the lowest memory usage (Mb) among all baselines and a shorter runtime per episode (s) compared to competitive non-stationary RL methods such as SeCBAD and COREP. WISDOM is also more efficient than the classical meta-RL method PEARL, despite PEARL not modeling task changes.
> >
> > **Table 6: Computational cost comparison among different methods.**
> >
> > |Task|WISDOM|PEARL|SeCBAD|RL$^2$|TRIO|COREP|CEMRL|
> > |-|-|-|-|-|-|-|-|
> > |Button-Press|**432M**|455M|457M|459M|499M|695M|741M|
> > ||251.32s|255.96s|409.86s|185.84s|**170.66s**|1207.51s|171.44s|
> > |Walker-Vel|**406M**|421M|425M|449M|443M|501M|787M|
> > ||175.34s|183.21s|285.37s|**80.12s**|108.05s|348.94s|327.98s|
> >
> > **Q1**: Yes, our wavelet representation can be viewed as a plug-in module and can be connected with non-stationary policy methods. We conducted additional experiments to validate that WISDOM can be integrated with online/offline Non-Stationary RL (NSRL) methods as well as model-based RL. The testing curves on unseen tasks are provided in Appendix Fig.20 of the revised paper, and the final testing success rates are shown below:
> >
> > **Table 5: Final performance in evaluating the plug-in role of wavelet task representations in model-based RL and non-stationary online/offline setups.**
> >
> > |Task|DORA|DORA+WISDOM|LCPO|LCPO+WISDOM|MAMBA|MAMBA+WISDOM|
> > |-|-|-|-|-|-|-|
> > |Handle-Press|66.67$\pm$22.28|**84.67$\pm$10.70**|92.13$\pm$13.64|**99.90$\pm$0.20**|97.58$\pm$2.86|**99.58$\pm$0.59**|
> > |Door-Close|93.00$\pm$9.90|**98.83$\pm$1.65**|94.27$\pm$6.76|**98.83$\pm$1.64**|99.42$\pm$0.82|**99.92$\pm$0.12**|
> >
> > LCPO [1] is an online non-stationary policy learning method that does not incorporate additional representation learning; it mitigates catastrophic forgetting by constraining policy optimization using experience samples that lie outside the current context distribution. Building on LCPO, our wavelet-based task representation further accelerates adaptation and improves the final success rate. This further demonstrates the effectiveness of our wavelet representation and shows that it can facilitate policy improvement, as established in Theorem 3.
> >
> > DORA [2] is an offline NSRL method that incorporates mutual information to learn task representations capable of differentiating between changes in the environment and the behavior policy. By filtering out spurious high-frequency components and learning stable multi-scale features, our wavelet-based representations reduce misleading correlations in the data and mitigate the distribution shift, effectively preventing policy performance degradation, as observed in Door-Close in Fig.20.
> >
> > Using the model-based meta-RL method MAMBA [3], which is built on a world model, we show that incorporating our wavelet representation significantly improves adaptation efficiency on non-stationary tasks such as Handle-Press and Door-Close. The potential reason may be that the wavelet representation explicitly separates the non-stationary changing patterns, making it easier for the world model to capture the true environmental dynamics and thereby improve its prediction accuracy.
> >
> > [1] Pouya et al. Online Reinforcement Learning in Non-Stationary Context-Driven Environments. ICLR 2025.
> >
> > [2] Xinyu et al. Debiased Offline Representation Learning for Fast Online Adaptation in Non-stationary Dynamics. ICML 2024.
> >
> > [3] Rimon Z et al. Mamba: an effective world model approach for meta-reinforcement learning. ICLR 2024.

---

### Official Review · Reviewer_78sm · 2025-10-27

**Soundness:** 2
**Presentation:** 3
**Contribution:** 3
**Rating:** 4
**Confidence:** 3

**Summary:**

This paper proposes WISDOM, a method for Non-Stationary Reinforcement Learning (NSRL) that leverages wavelet-domain predictive task representations. The key idea is to perform a discrete wavelet transform (DWT) on latent task representations to capture both low-frequency trends and high-frequency variations in evolving MDPs. The authors introduce a wavelet temporal-difference update operator, theoretically prove its contraction property and convergence, and integrate it into a meta-RL framework for improved adaptability to dynamic environments. Experiments on Meta-World, MuJoCo, and Type-1 Diabetes benchmarks demonstrate improved sample efficiency and final performance compared to strong baselines such as PEARL, SeCBAD, CEMRL, and COREP.

**Strengths:**

1.The paper presents a creative use of wavelet decomposition for temporal abstraction in non-stationary task representation. Unlike conventional Fourier- or RNN-based modeling, the proposed approach explicitly encodes multi-scale temporal information.

2.The authors rigorously define the wavelet TD operator and prove its contraction mapping property (Theorem 1), establishing convergence guarantees rarely seen in representation-learning-based NSRL methods.

3.The appendices include detailed analyses on the effects of decomposition levels, TD loss coefficients, and architectural components, which substantiate the design choices.

**Weaknesses:**

1.While wavelet theory is well-motivated for time-series decomposition, the link between wavelet coefficients and latent task dynamics could be more clearly articulated.

2.The paper does not discuss the additional computational cost introduced by recursive wavelet decomposition and reconstruction. An empirical runtime comparison (e.g., training time per step or GPU hours) would make the work more practically relevant.

3.Although WISDOM outperforms baselines on the six reported Meta-World tasks, the evaluation scope remains narrow. Meta-World environments vary significantly in difficulty (e.g., reach vs. door-open vs. hammer) [1], and the paper only tests on relatively moderate-difficulty tasks. It remains unclear how WISDOM performs in harder manipulation environments or more complex, partially observable settings. Expanding the evaluation to a broader range of task difficulties would help establish the robustness and applicability of the proposed approach.

[1] Seo Y, Hafner D, Liu H, et al. Masked world models for visual control[C]//Conference on Robot Learning. PMLR, 2023: 1332-1344.

**Questions:**

1.How sensitive is WISDOM to the choice of the wavelet basis?

2.Does the wavelet TD update improve stability when the reward signal is extremely sparse, or mainly when the task transition distribution drifts?

3.Could the proposed representation be integrated with model-based RL frameworks (e.g. world model-based RL) to further leverage the predictive nature of wavelet transforms?

---

> ### Author Response · Authors · 2025-11-27
> **Responses 1/2**
>
> Thank you for your valuable comments, which have helped us improve the paper. We have revised the manuscript based on your feedback. The details of our revisions are provided below. Please let us know if these revisions address your questions.
>
> **W1: The link between wavelet coefficients and latent task dynamics**
>
> Fig.8 is intended to illustrate the connection between wavelet coefficients and latent task dynamics. We apologize that the small labels in Fig.8 may have made it hard to interpret. As shown in the middle subfigures of Fig.8, the approximation coefficients that represent low-frequency trends match the true underlying task changes and become smoother with deeper decomposition. In the right subfigures, the detail coefficients that represent high-frequency details show sharper, more localized changes and may appear inverted due to the properties of the wavelet basis. In the left subfigure, when we combine the final approximation with selected details, we obtain our wavelet task representation, which closely follows the true trend of the non-stationary task. Overall, this illustrates that the wavelet-based representation successfully captures the underlying structure of how tasks change over time.
>
> **W2: The recursive wavelet decomposition and reconstruction of WISDOM add only a modest computational cost.** Using the Button-Press and Walker-Vel tasks on Meta-World and MuJoCo as examples, WISDOM stands out with the lowest memory usage (Mb) among all baselines and a shorter runtime per episode (s) compared to competitive non-stationary RL methods such as SeCBAD and COREP. WISDOM is also more efficient than the classical meta-RL method PEARL, despite PEARL not modeling task changes.
>
> **Table 6: Computational cost comparison among different methods.**
>
> |Task|WISDOM|PEARL|SeCBAD|RL$^2$|TRIO|COREP|CEMRL|
> |-|-|-|-|-|-|-|-|
> |Button-Press|**432M**|455M|457M|459M|499M|695M|741M|
> ||251.32s|255.96s|409.86s|185.84s|**170.66s**|1207.51s|171.44s|
> |Walker-Vel|**406M**|421M|425M|449M|443M|501M|787M|
> ||175.34s|183.21s|285.37s|**80.12s**|108.05s|348.94s|327.98s|
>
> **W3: Additional experiments on suggested more difficult tasks**
>
> Thanks for your advice. We have included additional experiments on the more complex Reach, Door-Open, and Hammer tasks, as suggested by the reviewer. The testing curves on unseen tasks are provided in Appendix Fig.16 of the revised paper, and the final performance results are shown below:
>
> **Table 1: Converged average test success rate $\pm$ standard error (\%) on more difficult tasks.**
>
> |Task|WISDOM|COREP|SeCBAD|TRIO|CEMRL|
> |-|-|-|-|-|-|
> |Door-Open|**100.00$\pm$0.00**|33.00$\pm$44.64|0.00$\pm$0.00|0.00$\pm$0.00|0.00$\pm$0.00|
> |Hammer|**61.11**$\pm$42.23|6.67$\pm$9.22|0.00$\pm$0.00|0.00$\pm$0.00|2.33$\pm$**3.30**|
> |Reach|**28.00**$\pm$9.09|27.00$\pm$**8.12**|21.67$\pm$9.04|11.67$\pm$9.31|8.33$\pm$9.48|
>
> WISDOM consistently outperforms popular non-stationary baselines on these more challenging tasks, indicating that wavelet-based task representations can effectively capture the underlying trends of complex non-stationary changes and remain robust across a broader range of task difficulties.
>
> **Q1: Sensitivity analysis of wavelet basis.**
>
> We evaluated the sensitivity of WISDOM to the choice of wavelet basis by conducting additional experiments on Button-Press using three representative wavelet bases (Bior1.1, Sym2, and Haar). The results indicate that WISDOM is NOT sensitive to the choice of wavelet basis. As illustrated in the testing curves of Fig.17 in Appendix E.8 of our revised paper, different bases lead to slight differences in convergence speed, but they have minimal impact on the final performance. The final testing success rates $\pm$ standard errors (\%) are shown below:
>
> **Table 2: Final testing performance with different wavelet bases on Button-Press.**
>
> |Sym2|Bior1.3|Haar|
> |-|-|-|
> |**98.67$\pm$1.89**|98.44$\pm$2.20|97.78$\pm$3.14|
>
> In our wavelet representation network, the wavelet basis functions serve only as an initialization to approximate traditional wavelet behavior. Because the filters are learnable, the model progressively reduces reliance on the initial basis and is able to extract more expressive and task-adaptive features from non-stationary task sequences without relying on hand-crafted designs or a fixed wavelet basis. This explains why different bases influence convergence only in the early stages while having a minimal impact on the final performance. Please refer to Appendix E.8 for further details.

---

> ### Author Response · Authors · 2025-11-27
> **Responses 2/2**
>
> **Q2: Yes, the wavelet TD update improves stability in tasks with extremely sparse reward signals and transition distribution shifts.**
>
> In Fig.5 and Appendix Fig.15 of the original submission, we evaluated WISDOM on tasks with transition distribution shifts. The damping parameters in the Cheetah-Damping task (Fig.5) and the physical parameters of the agent, such as body mass, damping, and friction, in the Walker-Rand-Params task (Fig.15) change over time to simulate non-stationarity in the state transition dynamics. The testing results show that WISDOM adapts more rapidly than the baselines under such transition distribution shifts. We also conducted an additional experiment on Walker-Mass to further evaluate the wavelet TD update. The testing curves on unseen tasks are in Appendix Fig.19 of the revised paper, and the final results are shown below:
>
> **Table 4: Testing average return on Walker-Mass with transition distribution shifts.**
>
> |Task|WISDOM|w/o wavelet TD|CEMRL|TRIO|SeCBAD|COREP|
> |-|-|-|-|-|-|-|
> |Walker-Mass|**3688.35**$\pm$225.90|3471.55$\pm$475.02|3390.48$\pm$212.02|1652.98$\pm$488.82|1484.95$\pm$519.47|921.49$\pm$**54.62**|
>
> We also conducted additional experiments on tasks with sparse rewards. Following the sparse-reward setup in LatCo [1], the agent receives a reward only when it successfully completes the task. The testing curves on unseen tasks are provided in Appendix Fig.18 of the revised paper, and the final testing success rates are shown below:
>
> **Table 3: Final testing performance on tasks with sparse rewards.**
>
> |Task|WISDOM|w/o Wavelet TD|CEMRL|TRIO|SeCBAD|COREP|
> |-|-|-|-|-|-|-|
> |Sparse-Door-Close|**100.00$\pm$0.00**|99.78$\pm$0.31|46.89$\pm$42.63|0.00$\pm$0.00|0.00$\pm$0.00|0.00$\pm$0.00|
> |Sparse-Handle-Press|**100.00$\pm$0.00**|99.65$\pm$0.50|26.14$\pm$24.58|7.02$\pm$9.07|32.63$\pm$30.46|25.61$\pm$16.65|
>
> WISDOM consistently outperforms the baselines, and the wavelet TD update further enhances both adaptability and stability. This demonstrates that our wavelet representation can handle non-stationary changes not only in reward functions but also in state transition dynamics, enabling rapid adaptation. Importantly, the explicit wavelet TD update helps the learned representations become denser and more informative, while also highlighting low-reward yet critical features. Thus, WISDOM also exhibits exceptional adaptation efficiency on tasks with sparse rewards.
>
> **Q3: Additional experiment on integration with model-based RL frameworks.**
>
> Yes. Using the model-based meta-RL method MAMBA [2], which is built on a world model, we show that incorporating our wavelet representation significantly improves adaptation efficiency on non-stationary tasks such as Handle-Press and Door-Close. The potential reason may be that the wavelet representation explicitly separates the non-stationary changing patterns, making it easier for the world model to capture the true environmental dynamics and thereby improve its prediction accuracy.
>
> The testing curves on unseen tasks are provided in Appendix Fig.20 of the revised paper, and the final testing success rates are shown below:
>
> **Table 5: Final performance in evaluating the plug-in role of wavelet task representations in model-based RL and non-stationary online/offline setups.**
>
> |Task|DORA|DORA+WISDOM|LCPO|LCPO+WISDOM|MAMBA|MAMBA+WISDOM|
> |-|-|-|-|-|-|-|
> |Handle-Press|66.67$\pm$22.28|**84.67$\pm$10.70**|92.13$\pm$13.64|**99.90$\pm$0.20**|97.58$\pm$2.86|**99.58$\pm$0.59**|
> |Door-Close|93.00$\pm$9.90|**98.83$\pm$1.65**|94.27$\pm$6.76|**98.83$\pm$1.64**|99.42$\pm$0.82|**99.92$\pm$0.12**|
>
> Besides, we also conducted an additional experiment to validate that our wavelet representation can be integrated with other online non-stationary policy learning methods, such as LCPO [3], as well as with offline methods such as DORA [4]. When combined with these methods, our wavelet-based task representation further accelerates adaptation and improves the final success rate. This further validates the generality and effectiveness of our wavelet representation and shows that it can facilitate policy improvement, consistent with our analysis in Theorem 3.
>
> [1] Oleh et al. Model-Based Reinforcement Learning via Latent-Space Collocation. ICML 2021
>
> [2] Rimon Z et al. Mamba: an effective world model approach for meta-reinforcement learning. ICLR 2024.
>
> [3] Pouya et al. Online Reinforcement Learning in Non-Stationary Context-Driven Environments. ICLR 2025.
>
> [4] Xinyu et al. Debiased Offline Representation Learning for Fast Online Adaptation in Non-stationary Dynamics. ICML 2024.

---

### Official Review · Reviewer_8h4R · 2025-10-28

**Soundness:** 3
**Presentation:** 3
**Contribution:** 3
**Rating:** 8
**Confidence:** 4

**Summary:**

This paper introduces a novel reinforcement learning (RL) method for non-stationary domains. In particular, it is considered the setting in which the agent interacts with a sequence of MPDs, and the evolution of such MPDs is determined by a history-dependent stochastic process. To tackle this setting, the authors propose to track task  evolution via a Wavelet representation network. The proposed method, named WISDOM, is based on a Soft Actor-Critic (SAC) agent that also learns a wavelet representation network, which predicts a vector representation of the current task, and is optimized via a TD-learning loss combined with an auto-regressive loss function. The paper introduces two theorems that characterize policy performance with respect to wavelet-domain features. The experiments considered Meta-World, Type-1 Diabetes, and MuJoCo benchmarks, and compared the proposed approach with many competing methods in non-stationary RL.

**Strengths:**

- The idea of using a Wavelet representation to track non-stationary in RL is interesting and novel.
- The experiments are extensive, considered challenging tasks, and have evaluated many competing methods. Moreover,  the authors provided the source code for reproducing the experiments.

**Weaknesses:**

- The paper can benefit from a more rigorous/precise mathematical notation in some parts of the text (see Questions below).
- The description and discussion of Theorem 3 require significant improvements in terms of clarity. For instance, the variable $J_{WISDOM}$ is not defined, nor is it clearer what is meant by $its iterated history policy$.

**Questions:**

- Section 3.2 would benefit from more details on the problem formulation. In particular, I suggest discussing the intuition behind the formulation in Eq. 1. For instance, $\mathcal{T}$ is a set of tasks, and $p(\mathcal{T})$ is a distribution of tasks. Then, the authors introduce $p(\mathcal{T_t})$, whose definition is not very clear. In Eq. 1., it is not clear how an expected value over $\mathcal{T}_t)$ relates to the expected value over $\omega_h$. Providing an example of a real-world scenario where Eq. 1 makes sense would improve the clarity of this section.

- In line 257, $\Gamma$ is not defined. Why does this operator not have an expected value over $z_{t+1}$? Moreover, why does this operator not depend on the policy being followed?

- How does the policy being followed impact the Wavelet representations? How does the model differ from non-stationarity of the task, to non-stationarity caused by policy being updated and changing

- “Stabilize the training and is updated separately and softly.” Please define what is meant by “softly”. Did you mean with polyak averaging?

- In Theorems 2 and 3, the variable $J_\pi$ was not defined. Is it the performance in a single episode? An average over the entire space of tasks? Please be precise.

- There is also a class of non-stationary RL methods that track context changes in the MDP, which could have been at least discussed in the related work. See [1-3].

[1] Silva, Bruno C. da, Eduardo W. Basso, Ana L. C. Bazzan, and Paulo M. Engel. ‘Dealing with Non-Stationary Environments Using Context Detection’. Proceedings of the 23rd International Conference on Machine Learning, 2006.

[2] Alegre, Lucas N., Ana L. C. Bazzan, and Bruno C. da Silva. ‘Minimum-Delay Adaptation in Non-Stationary Reinforcement Learning via Online High-Confidence Change-Point Detection’. Proceedings of the 20th International Conference on Autonomous Agents and MultiAgent Systems, 2021.

[3] Feng, Fan, Biwei Huang, Kun Zhang, and Sara Magliacane. ‘Factored Adaptation for Non-Stationary Reinforcement Learning’. Advances in Neural Information Processing Systems 35, 2022.

---

> ### Author Response · Authors · 2025-11-27
> **Responses 1/2**
>
> Thank you for reviewing our paper and providing valuable comments. Below we respond to each of the questions you raised and have revised the manuscript accordingly. We would greatly appreciate it if you could let us know whether our responses adequately address your concerns.
>
> **W1**: Thanks for your advice. We have revised the relevant mathematical notations in response to the following questions and have highlighted the changes in blue in the updated submission.
>
> **W2**: Thanks for pointing it out. $J_\text{WISDOM}$ denotes the performance of WISDOM’s final policy. The iterated history policy $\pi_h$ refers to the intermediate policy produced during WISDOM’s policy iteration process. We have defined these variables in the revised version for clarity.
>
> **Q1**: Thank you for the suggestion. We added the following example to clarify Eq.(1).
>
> Consider a Walker task in MuJoCo where the agent’s mass changes twice during interaction, resulting in three distinct transition dynamics corresponding to MDPs $\mathcal{\omega}_0$, $\mathcal{\omega}_1$, and $\mathcal{\omega}_2$. Their durations are $T_0$, $T_1$, and $T_2$, with $h\in\{0,1,2\}$ and timestep $t$ ranging from 0 to $T_0+T_1+T_2-1$. The goal of non-stationary RL is to find a policy that maximizes the expected cumulative reward over this entire trajectory, with a horizon of $T_0+T_1+T_2$. In general, $p(\mathcal{T})$ represents the distribution of stationary tasks, whereas in our non-stationary setup, we use $p(\mathcal{T}_t)$ to denote the distribution of time-evolving tasks. We have incorporated this explanation and example into the revised version to clarify the intuition behind Eq.(1).
>
> **Q2**: Thank you for pointing it out. $\Gamma$ is defined in Theorem 1 (line 294) as the discount factor in diagonal matrix form, and we have further clarified this in line 273 of the revised paper. Consistent with Eq.(5), the $\Gamma$ operator should also include the expectation over $z_{t+1}$. And this operator also depends on the policy being followed. We have corrected this in the revised version.
>
> **Q3**: Thank you for raising this important question. In the long run, the policy being followed has little impact on the wavelet representations, and this effect is generally ignored in non-stationary MDP setups with downstream policy learning methods. Our WISDOM is based on the Soft Actor-Critic (SAC) algorithm, which mitigates the non-stationarity caused by policy updates through the use of a replay buffer and target network updates, similar to Deep Q-Networks (DQN).
>
> Non-stationary tasks/MDPs typically refer to shifts in the reward or transition dynamics. In non-stationary RL, such task changes are often assumed to follow certain trends or periodic patterns, whereas the non-stationarity introduced by policy updates is generally random. Consistent with context-based meta-RL, the input task representation of the wavelet representation network is obtained by encoding trajectories. Since the policy being followed may impact the action distribution and the state-action visitation, policy updates potentially introduce noise into the task representations. Our wavelet representation network helps distinguish between the two types of non-stationarity: the approximation coefficients capture the global trends associated with task changes, while downsampling filters out some detail coefficients that represent high-frequency noise caused by policy-induced non-stationarity. Thus, the resulting wavelet representations reflect the underlying task changes.
>
> **Q4**: Yes, following the "soft target update" used in DDPG [4], "softly" in our paper means updating with Polyak averaging. Specifically, our target network $W_\mu$ is updated using an exponential moving average, a variant of Polyak averaging, of the parameters of the $W_\varphi$ network, similar to the target Q-network update in DDPG. We defined this in Appendix C (line 1158) and have further clarified it in the revised version.
>
> [4] Timothy et al. Continuous control with deep reinforcement learning. ICLR 2026.
>
> **Q5**: Yes, your conjecture is correct. We have defined $J_\pi$ in the revised version for better readability.

---

> > ### Author Response · Authors · 2025-11-27
> > **Responses 2/2**
> >
> > **Q6**: Thank you for pointing out these relevant works. We have incorporated the recommended references into the revised Related Work section. [1] and [2] are both non-stationary change detection methods. [1] utilizes a set of partial models, each specialized for a different environment, to predict environmental dynamics. The partial model with the lowest prediction error is activated, and a non-stationary change is detected whenever the active model switches. [2] builds an ensemble of context-dynamics predictors to capture different modes of the underlying task distribution. New predictors are added to the ensemble when entirely new contexts are detected for the first time. Similar to our baseline COREP [ICML 2024], [3] also models the transition and reward functions using a causal graph to handle non-stationary settings, including continuous and discrete changes. In contrast, COREP employs a dual-graph structure to separately learn stable representations and detailed edge information.
> >
> > Different from them, we first introduce a principled mechanism to track and adapt to task dynamics in the wavelet domain, rather than relying entirely on implicit learning from data. Through iterative wavelet decomposition, WISDOM naturally separates evolving patterns across multiple frequencies, enabling it to capture both fast and slow task variations without requiring pattern switching, additional predictors, or computationally expensive causal-graph modeling (as in [3]). As a result, WISDOM is substantially more efficient, and the resulting multi-scale wavelet task representations more faithfully reflect the underlying structural and temporal patterns of non-stationary tasks.

---

### Official Review · Reviewer_hSXg · 2025-11-01

**Soundness:** 3
**Presentation:** 3
**Contribution:** 3
**Rating:** 6
**Confidence:** 3

**Summary:**

This paper introduces WISDOM, a context-based non-stationary RL that models sequences of latent task representations in the wavelet domain. A learnable wavelet representation network, composed of two dilated causal convolution layers functioning as adaptive low- and high-pass filters, decomposes context embeddings into approximation and detail coefficients, selectively downsamples detail terms, and reconstructs an enhanced representation used by the policy and critic.
Training jointly applies a contraction-guaranteed wavelet TD operator and an autoregressive likelihood objective. The theoretical analysis shows that wavelet-domain features can bound policy performance differences and that reconstructed representations improve contextual policy learning under certain assumptions.

**Strengths:**

- The paper evaluates non-stationary RL scenarios across Meta-World, MuJoCo, and Type-1 Diabetes environments. The inclusion of multiple benchmarks and difficulty levels provides a strong empirical basis for assessing adaptability and generalization.

- WISDOM shows particularly superior performance on Meta-World, while achieving comparable or better results in MuJoCo and other environments, supporting the claimed contributions.

- The paper is easy to read and follow.

**Weaknesses:**

- Discussion is lacking on how the proposed wavelet-based approach for tracking fast and slow changes connects to prior methods. This omission makes the contribution somewhat unclear; it is not well explained why alternative approaches could not be applied or how the proposed one provides more than an implementation variant.

- Among the baselines, COREP also models non-stationarity by disentangling stable and varying causal factors through a dual-graph structure, conceptually related to WISDOM's multi-scale decomposition. A clearer discussion on why WISDOM's frequency-based modeling achieves stronger results than COREP would further strengthen the paper.

- To further enhance reproducibility, it would be helpful to include additional details on how hyperparameter tuning was performed for each baseline, particularly in environments not examined in their original studies. In the same context, offering clarification on why CEMRL exhibits lower performance on Meta-World compared to other methods would further enrich the empirical analysis and provide valuable insights for replication and interpretation.

**Questions:**

- There is an interesting sharp performance increase pattern around 4e5 time steps in Figure 3 (Meta-World experiments) for WISDOM (except Plate-Slide-Back). Likewise, in Figure 5 (MuJoCo experiments) around 0.2 million steps, all baselines show a sudden performance improvement. What factors cause these abrupt jumps in performance?

---

> ### Author Response · Authors · 2025-11-27
> **Responses 1/2**
>
> Thank you for your insightful comments. We have carefully considered each of your suggestions and revised the paper accordingly. Our detailed responses to your comments are provided below. We would appreciate it if you could inform us whether our revisions and explanations adequately address your concerns.
>
> **W1: The superiority of wavelet-based WISDOM for tracking fast and slow changes compared to prior methods**
>
> We would like to clarify that our WISDOM is not an implementation variant of previous Non-Stationary Reinforcement Learning (NSRL) methods. To the best of our knowledge, WISDOM is the first to address NSRL by leveraging latent representations to capture the underlying task dynamics from the perspective of frequency transformation. This insight is also conceptually supported by wavelet theory. WISDOM explicitly incorporates a modeling bias that aligns with the underlying temporal structure of non-stationary environments. We introduce a principled mechanism to track and adapt to task dynamics in the wavelet domain, rather than relying entirely on implicit learning from data, as prior methods do. Moreover, we design a wavelet TD update explicitly applied to the representation, along with the auto-regressive (AR) loss, to enhance prediction accuracy and highlight low-reward yet critical features. These are particularly beneficial in data-scarce settings, where purely end-to-end models may struggle to capture evolving patterns due to limited signals.
>
> Most prior NSRL methods detect task changes through reward functions, such as SeCBAD and the Gaussian Mixture Model (GMM)-based approach in CEMRL, so they struggle to track both fast and slow changes. Although COREP models non-stationarity by disentangling stable and varying causal factors through a dual-graph structure, this binary disentanglement is relatively coarse and cannot effectively distinguish changes occurring at different frequencies. TRIO instead models task evolution as a Gaussian Process (GP). While non-stationary GP kernels can capture frequency differences, they require prior knowledge for kernel selection and substantially increase model complexity.
>
> In contrast, we adopt a time-series perspective and treat the evolving task sequence as a non-stationary signal, whose underlying frequency changes over time. Our wavelet representation network iteratively decomposes the original task representation into two components: 1) low-frequency approximation coefficients that capture slow, long-term task trends, and 2) high-frequency detail coefficients that reflect fast or noisy changes. With each decomposition, more high-frequency noise is removed, yielding a smoother global trend. After several iterations, the final approximation coefficients encode the global pattern of task evolution, while selected detail coefficients preserve informative local variations. Moreover, our learnable filters reduce reliance on specific wavelet bases, allowing for more expressive and task-adaptive feature extraction without hand-crafted design.
>
> We have added new experiments on more complex tasks, including Reach, Door-Open, Hammer, and sparse-reward tasks. WISDOM consistently outperforms all baselines and achieves significantly higher final performance. The testing curves on unseen tasks are reported in Appendix Fig.16 and Fig.18 of the revised paper, and the corresponding final success rates are shown below:
>
> **Table 1: Converged average test success rate $\pm$ standard error (\%) on more difficult tasks.**
>
> |Task|WISDOM|COREP|SeCBAD|TRIO|CEMRL|
> |-|-|-|-|-|-|
> |Door-Open|**100.00$\pm$0.00**|33.00$\pm$44.64|0.00$\pm$0.00|0.00$\pm$0.00|0.00$\pm$0.00|
> |Hammer|**61.11**$\pm$42.23|6.67$\pm$9.22|0.00$\pm$0.00|0.00$\pm$0.00|2.33$\pm$**3.30**|
> |Reach|**28.00**$\pm$9.09|27.00$\pm$**8.12**|21.67$\pm$9.04|11.67$\pm$9.31|8.33$\pm$9.48|
>
> **Table 3: Final testing performance on tasks with sparse rewards.**
>
> |Task|WISDOM|w/o Wavelet TD|CEMRL|TRIO|SeCBAD|COREP|
> |-|-|-|-|-|-|-|
> |Sparse-Door-Close|**100.00$\pm$0.00**|99.78$\pm$0.31|46.89$\pm$42.63|0.00$\pm$0.00|0.00$\pm$0.00|0.00$\pm$0.00|
> |Sparse-Handle-Press|**100.00$\pm$0.00**|99.65$\pm$0.50|26.14$\pm$24.58|7.02$\pm$9.07|32.63$\pm$30.46|25.61$\pm$16.65|

---

> > ### Author Response · Authors · 2025-11-27
> > **Responses 2/2**
> >
> > **W2: The reason why WISDOM's frequency-based modeling achieves stronger results than COREP**
> >
> > The key difference is that COREP’s notion of "stable vs. varying" factors reflects a spatial structural separation, whereas WISDOM’s wavelet-based "slow vs. fast changes" represents a time–frequency signal decomposition. Specifically, COREP learns two graph structures: the core graph, which captures stable and essential relationships (e.g., inherent object composition), and the general graph, which captures instance-specific, rapidly changing details (e.g., lighting or viewpoint). This factorization focuses on representation learning to improve stability and generalization. In contrast, the wavelet transform in WISDOM operates purely at the signal-processing level, expanding a temporal signal into the time–frequency domain using mathematical kernels: low-frequency components reflect global trends and slow changes, while high-frequency components capture fast changes or noise without learning any hand-crafted or learned task structures. As a result, COREP’s stable/variable factorization is a relatively coarse, binary partition that captures semantic or causal stability but cannot provide the fine-grained, multi-resolution characterization of non-stationarity that WISDOM achieves through its frequency-based modeling.
> >
> > **W3**: Thanks for your advice. For fair comparisons, we adopt the hyperparameter settings reported in the original papers for the environments included in their studies. For environments not covered in the original works, we tune the hyperparameters within the ranges specified by the authors' open source implementations to ensure strong performance.
> >
> > **Clarification on the performance of CEMRL on Meta-World**
> >
> > Although CEMRL exhibits higher performance in MuJoCo compared to other baselines, it struggles significantly in more challenging benchmarks, such as Meta-World. This substantial performance gap of CEMRL stems from differences in task distributions and inherent limitations in CEMRL’s task inference module.
> >
> > Compared to the classical MuJoCo setting, Meta-World features a much broader task distribution, making it substantially more difficult. As noted in the Meta-World paper [Yu et al., CoRL 2019], “previous evaluation benchmarks utilize very narrow task distributions, making it difficult to understand the degree to which meta-RL actually enables generalization.”
> >
> > CEMRL models non-stationary task distributions with a Gaussian Mixture Model (GMM) to cluster task representations. When the task distribution is narrow, GMM can easily cluster tasks into a few limited categories. However, as the task distribution widens, increasing the number of clusters dramatically raises training cost, while reducing the number of clusters risks representation collapse and poor task inference. By contrast, our WISDOM performs consistently well on Meta-World, demonstrating its strong robustness and rapid adaptability.
> >
> > **Q1**: Thank you for the thoughtful observation. We observe that Figures 3 and 5 exhibit sharp increases in performance after a certain number of training steps. Similar behavior has also been reported in recent RL literature [1]. It also reports that model performance can undergo a sudden, phase-transition–like leap after tens of thousands of training steps. They refer to this phenomenon as "RL grokking", highlighting that true learning often emerges when chaotic trial-and-error transfers into structured understanding.
> >
> > In our setting, the performance curves in Figures 3 and 5 are measured on unseen test tasks to assess adaptability. The abrupt performance jumps tend to be consistent with this RL grokking effect, indicating that WISDOM is not memorizing task-specific solutions but is instead internalizing the underlying rules of non-stationary changes after substantial trial-and-error. The testing performance appears to be aligned with human behavior when tackling completely new problems: blind attempts in the early stage, key breakthroughs in the middle stage, and rapid convergence in the later stage.
> >
> > We believe that the grokking effect emerges earlier and more prominently for WISDOM due to its wavelet-based representation-learning design. As explained in our response to W1, WISDOM’s multi-scale wavelet decomposition extracts more intrinsic wavelet task representations that capture the structural information of non-stationary environments. The wavelet TD loss further guides the representation network to learn regularities and improve sample efficiency, thereby reducing the amount of blind exploration. Additionally, the AR loss helps predict global trends in task evolution, which are essential for learning a generalizable policy across diverse non-stationary tasks. Together, these components accelerate the emergence of the grokking moment.
> >
> > [1] Sun Y, Cao Y, Huang P, et al. RL Grokking Recipe: How Does RL Unlock and Transfer New Algorithms in LLMs?[J]. arXiv preprint arXiv:2509.21016, 2025.

---

### Author Response · Authors · 2025-11-27
**Common Response**

We sincerely appreciate the time and effort that the area chair and all the reviewers devoted to reviewing this paper. We have responded to each weakness and question raised by the reviewers and revised the manuscript accordingly. All changes are highlighted in blue in the updated submission. In addition, we conducted further experiments to more comprehensively evaluate our proposed method. **The corresponding training curves and details are provided in Appendix Figures 16–21.** The final performance results of these new experiments are summarized below:

**Table 1: Converged average test success rate $\pm$ standard error (\%) on more difficult tasks.**

|Task|WISDOM|COREP|SeCBAD|TRIO|CEMRL|
|-|-|-|-|-|-|
|Door-Open|**100.00$\pm$0.00**|33.00$\pm$44.64|0.00$\pm$0.00|0.00$\pm$0.00|0.00$\pm$0.00|
|Hammer|**61.11**$\pm$42.23|6.67$\pm$9.22|0.00$\pm$0.00|0.00$\pm$0.00|2.33$\pm$**3.30**|
|Reach|**28.00**$\pm$9.09|27.00$\pm$**8.12**|21.67$\pm$9.04|11.67$\pm$9.31|8.33$\pm$9.48|

**Table 2: Final testing performance with different wavelet bases on Button-Press.**

|Sym2|Bior1.3|Haar|
|-|-|-|
|**98.67$\pm$1.89**|98.44$\pm$2.20|97.78$\pm$3.14|

**Table 3: Final testing performance on tasks with sparse rewards.**

|Task|WISDOM|w/o Wavelet TD|CEMRL|TRIO|SeCBAD|COREP|
|-|-|-|-|-|-|-|
|Sparse-Door-Close|**100.00$\pm$0.00**|99.78$\pm$0.31|46.89$\pm$42.63|0.00$\pm$0.00|0.00$\pm$0.00|0.00$\pm$0.00|
|Sparse-Handle-Press|**100.00$\pm$0.00**|99.65$\pm$0.50|26.14$\pm$24.58|7.02$\pm$9.07|32.63$\pm$30.46|25.61$\pm$16.65|

**Table 4: Testing average return on Walker-Mass with transition distribution shifts.**

|Task|WISDOM|w/o wavelet TD|CEMRL|TRIO|SeCBAD|COREP|
|-|-|-|-|-|-|-|
|Walker-Mass|**3688.35**$\pm$225.90|3471.55$\pm$475.02|3390.48$\pm$212.02|1652.98$\pm$488.82|1484.95$\pm$519.47|921.49$\pm$**54.62**|

**Table 5: Final performance in evaluating the plug-in role of wavelet task representations in model-based RL and non-stationary online/offline setups.**

|Task|DORA|DORA+WISDOM|LCPO|LCPO+WISDOM|MAMBA|MAMBA+WISDOM|
|-|-|-|-|-|-|-|
|Handle-Press|66.67$\pm$22.28|**84.67$\pm$10.70**|92.13$\pm$13.64|**99.90$\pm$0.20**|97.58$\pm$2.86|**99.58$\pm$0.59**|
|Door-Close|93.00$\pm$9.90|**98.83$\pm$1.65**|94.27$\pm$6.76|**98.83$\pm$1.64**|99.42$\pm$0.82|**99.92$\pm$0.12**|

---

### Meta-Review · Area_Chair_uZhQ · 2026-01-04

**Summary:**

I suggest an accept based on the fact that the reviewers appreciated the novelty, theoretical support, and extensive experiments. The main concerns by the reviewers are also addressed as I have mentioned below.

I strongly suggest adding GPU hours or per-step training time.

**Reviewer Concerns:**

Most concerns received concrete rebuttal actions.

Reviewer hSXg was concerned about connection to previous methods, e.g., whether it is an implementation variant of a previous method. The authors clarified that it is not.

Reviewer 78sm sought connection to latent task dynamics, mention of additional computational cost, and performance on harder variants of meta-world tasks. The authors mentioned that the connection is provided, they provided a table with wall clock times, and added additional complex tasks.

Reviewer SUaC sought analysis of the learned representation, e.g., its quality, to understand whether the performance benefit is coming from the proposed wavelet-domain representation. The authors responded that they provided analysis of the learned representation.

**Reviewer Scores:**

Reviewer hSXg, 78sm, and SUaC would have likely increased their scores. The other reviewer’s score is already high.

---

### Decision · Program_Chairs · 2026-01-26

Accept (Poster)